# The Dual Faces of Oestrogen: The Impact of Exogenous Oestrogen on the Physiological and Pathophysiological Functions of Tissues and Organs

**DOI:** 10.3390/ijms25158167

**Published:** 2024-07-26

**Authors:** Joanna Bartkowiak-Wieczorek, Agnieszka Jaros, Anna Gajdzińska, Paulina Wojtyła-Buciora, Igor Szymański, Julian Szymaniak, Wojciech Janusz, Iga Walczak, Gabriela Jonaszka, Agnieszka Bienert

**Affiliations:** 1Physiology Department, Poznan University of Medical Sciences, 61-701 Poznan, Poland; 89064@student.ump.edu.pl (A.G.); paulinawotyla@ump.edu.pl (P.W.-B.); 88856@student.ump.edu.pl (I.S.); 88858@student.ump.edu.pl (J.S.); 87712@student.ump.edu.pl (W.J.); 90155@student.ump.edu.pl (I.W.); 79544@student.ump.edu.pl (G.J.); 2Department of Clinical Pharmacy and Biopharmacy, Poznan University of Medical Sciences, 61-701 Poznan, Poland; agnieszkajaros0@gmail.com (A.J.); agbienert@ump.edu.pl (A.B.); 3Department of Social Medicine and Public Health, Calisia University, 62-800 Kalisz, Poland

**Keywords:** oestrogen, phytoestrogen, xenoestrogen, oestrogen-dependent cancer, source, environmental exposition

## Abstract

Oestrogen plays a crucial physiological role in both women and men. It regulates reproductive functions and maintains various non-reproductive tissues through its receptors, such as oestrogen receptor 1/oestrogen receptor α (ESR1/Erα), oestrogen receptor 2/oestrogen receptor β (ESR2/Erβ), and G protein-coupled oestrogen receptor 1 (GPER). This hormone is essential for the proper functioning of women’s ovaries and uterus. Oestrogen supports testicular function and spermatogenesis in men and contributes to bone density, cardiovascular health, and metabolic processes in both sexes. Nuclear receptors Er-α and Er-β belong to the group of transcription activators that stimulate cell proliferation. In the environment, compounds similar in structure to the oestrogens compete with endogenous hormones for binding sites to receptors and to disrupt homeostasis. The lack of balance in oestrogen levels can lead to infertility, cancer, immunological disorders, and other conditions. Exogenous endocrine-active compounds, such as bisphenol A (BPA), phthalates, and organic phosphoric acid esters, can disrupt signalling pathways responsible for cell division and apoptosis processes. The metabolism of oestrogen and its structurally similar compounds can produce carcinogenic substances. It can also stimulate the growth of cancer cells by regulating genes crucial for cell proliferation and cell cycle progression, with long-term elevated levels linked to hormone-dependent cancers such as breast cancer. Oestrogens can also affect markers of immunological activation and contribute to the development of autoimmune diseases. Hormone replacement therapy, oral contraception, in vitro fertilisation stimulation, and hormonal stimulation of transgender people can increase the risk of breast cancer. Cortisol, similar in structure to oestrogen, can serve as a biomarker associated with the risk of developing breast cancer. The aim of this review is to analyse the sources of oestrogens and their effects on the endogenous and exogenous process of homeostasis.

## 1. Introduction

Oestrogens belong to the group of steroid hormones and exert regulatory control over a wide array of physiological processes in both women and men.

The biosynthesis of natural oestrogens involves the conversion of androgens (such as testosterone and androstenedione) into various oestrogenic compounds through aromatisation, yielding oestradiol, oestrone, and oestriol. This process primarily occurs in premenopausal women’s ovaries and in adipose tissue, adrenal glands, skin, the brain, the liver, and placenta [1].

Two main subtypes of oestrogen receptors, oestrogen receptor 1/oestrogen receptor α (ESR1/Erα) and oestrogen receptor 2/oestrogen receptor β (ESR2/Erβ), have been identified, playing pivotal roles in modulating the physiological functions of oestrogenic compounds. Despite often being labelled female hormones, oestrogens are indispensable for proper male organism function [2]. They contribute significantly to physiological processes such as preparing the uterine lining for embryo implantation, milk secretion during lactation, the maintenance of pregnancy, and bone development during adolescence [3].

Exogenous compounds that mimic oestrogen, activate receptor pathways, or interfere with regular oestrogenic activity are termed xenoestrogens [4]. Sources of xenoestrogens include plastics, pesticides, exhaust fumes, tobacco smoke, cosmetics, and certain plants, contributing to continuous human exposure to their effects [5].

Phytoestrogens, plant compounds structurally resembling 17-β oestradiol, exert estrogenic effects due to their similarity. Primary dietary sources of phytoestrogens include isoflavones, prenylflavonoids, coumestans, and lignans [6]. Phytoestrogens are present in various human food sources, such as fruits (plums, pears, apples, grapes, and berries), vegetables (beans, sprouts, cabbage, spinach, and soybeans), grains, hops, garlic, onions, and beverages like wine and tea [7].

Disruption in endogenous oestrogen synthesis underlies the pathophysiology of many disorders, necessitating external sources for treatment. FDA-approved indications for oestrogen use include primary ovarian insufficiency, female hypogonadism, symptoms of menopause (e.g., vulvovaginal atrophy, dyspareunia, hot flashes, and night sweats), prevention of osteoporosis, oral contraception, and management of moderate acne vulgaris and advanced metastatic prostate cancer [3].

Both deficiency and excess oestrogen can lead to health complications and disruptions in bodily functions, emphasising the importance of maintaining optimal oestrogen levels. Oestrogen’s multifaceted roles implicate it in the pathophysiology of various diseases and conditions, including infertility, obesity, osteoporosis, endometriosis, and several cancers, such as breast, ovarian, and endometrial cancers [8].

The regulation of oestrogen levels is complex, influenced by endogenous processes and numerous external factors. While dietary choices can be controlled, factors like air quality and water contamination remain beyond individual influence.

This review aims to identify sources of oestrogen, both endogenous and exogenous, focusing on their uncontrolled impacts and adverse effects while also comparing these with the beneficial aspects of oestrogen to provide a dual perspective on the hormone.

## 2. Oestrogen Characteristics (Type of Oestrogens, Synthesis, and Production)

Oestrogens are steroid hormones primarily responsible for stimulating and controlling the development and function of reproductive organs [9]. Oestrogens are regulated in various extragonadal functions, expressing their activity in the liver, heart, muscles, bones, and brain [10]. The synthesis and signalling of oestrogens can be tissue- and cell-specific [11]. In premenopausal women, oestrogens are mainly produced in the thecal cells of the ovaries, corpus luteum, and placenta, with lesser contributions from the liver, heart, skin, and brain. After menopause, adrenal glands and adipose tissue take over the role of oestrogen synthesis [12]. In men, these hormones are synthesised primarily in the testes, Sertoli cells, Leydig cells, and spermatocytes [13].

There are four main types of physiological oestrogens in women: oestrone (E1), oestradiol (E2 or 17β-oestradiol), oestriol (E3), and oestetrol (E4). These hormones differ in their potency and physiological effects. As noted in Coelingh Bennink et al., each oestrogen type has distinct roles: E2 induces proliferative activity, causing the growth of granulosa cells in ovarian follicles, determining and maintaining female sex characteristics, and controlling the reproductive cycle, pregnancy, embryonic, and foetal development [14]. Disruption of E2 homeostasis contributes to the development of breast, endometrial, and ovarian cancers. E2 also regulates liver and adipose metabolism and bone metabolism by inhibiting osteoclast formation, directly stimulating osteoblast activity and their differentiation from mesenchymal cells; calcitonin secretion increases under E2. Oestradiol deficiency contributes to osteoporosis and Alzheimer’s disease (AD) by regulating neuronal activity and contributes to arteriosclerosis by regulating vascular activity. E1 has a role similar to E2 but is 5–10 times less potent. At the same time, E3 effectively controls early menopausal symptoms, lacks proliferative effects (on the uterus and breast gland), and influences only the development of external genital organs [15].

Oestrogen metabolism plays a crucial role in regulating the levels of these hormones. E2 can be converted into less-active forms like E1 or E3, or 17beta-estra-1,3,5-trien-3,17-diol 3-sulfate through sulfation by oestrogen sulfotransferase, rendering it inactive for binding to oestrogen receptors. The balance between oestrogen synthesis and deactivation can be reflected in the ratio of circulating oestrogens, providing insights into dynamic metabolic processes. Aromatase, also known as cytochrome P450 19A1 (*CYP19A1*), is crucial for synthesising oestrogens from androgens in both women and men [16]. Regulation of aromatase enzyme activity is critical for controlling oestrogen synthesis in the body and maintaining various physiological functions across different life stages [15].

Aromatase catalyses the conversion of androgens to oestrogens and is widely expressed in the gonads of both sexes. In female ovaries, aromatase is primarily expressed in granulosa and luteal cells, whereas in male gonads, it is expressed in testes and accessory glands. This distribution ensures adequate oestradiol levels for normal spermiogenesis, sperm maturation, and sperm motility [17].

Studies indicate an age-related increase in aromatase activity in subcutaneous adipose stromal cells and aromatase mRNA levels in adipose tissue [18], potentially impacting oestrogen homeostasis. 

Aromatase expression is tightly regulated by tissue-specific promoters, allowing for selective modulation of oestrogen production in various tissues, including the placenta, gonads, adipose tissue, osteoblasts, and bone chondrocytes [19]. The enzyme is also widely distributed in the brain, where reactive astroglial forms express it in response to injury and disease [20].

Aromatase regulation involves multiple mechanisms. In the ovaries, follicle-stimulating hormones (FSHs) play a primary role in regulating aromatase gene expression. In the placenta, aromatase expression responds to elevated oestrogen levels during pregnancy. In adipose tissue and bones, glucocorticoids, class 1 cytokines, and tumour necrosis factor α (TNFα) regulate aromatase gene expression [21] (Table 1).

Beyond transcriptional control, aromatase activity can be influenced by post-translational modifications such as phosphorylation. Aromatase activity is notably inhibited by increased ATP, Mg^2+^, or Ca^2+^ concentrations, depending on protein kinase activity. Genistein, a tyrosine kinase inhibitor, and staurosporine, a serine/threonine kinase inhibitor, can effectively block ATP-, Mg^2+^-, or Ca^2+^-induced inhibition, suggesting a role for phosphorylation in regulating aromatase activity [19].

Various factors regulate aromatase expression, including hormones, cytokines, growth factors, and local signals. The aromatase gene (*CYP19A1*) contains multiple promoters, enabling tissue-specific regulation. For example, luteinising hormones (LHs) stimulate aromatase expression in the ovaries, whereas insulin and cytokines play regulatory roles in adipose tissue. In premenopausal women, ovarian aromatase is pivotal for oestrogen synthesis, converting androgens from theca cells into oestrogens in the granulosa cells of ovarian follicles. Post menopause, adipose tissue aromatase becomes crucial for maintaining oestrogen levels. In the placenta, aromatase synthesises oestrogens essential for pregnancy maintenance and foetal development. Additionally, local aromatase in the brain, bones, muscles, and skin facilitates the synthesis of oestrogens for paracrine and autocrine functions [22] (Table 1).

## 3. Oestrogen Receptors

Oestrogens act through two types of receptors: classical nuclear receptors (ER-α and ER-β) and cell surface receptors (GPR30 and ER-X). Oestrogen receptors (ERs) belong to the superfamily of ligand-induced nuclear receptors, including steroid, thyroid, and retinoic acid receptors, which regulate gene expression through protein–DNA or protein–protein interactions with other transcription factors [23]. Two types of oestrogen receptors, ER-α and ER-β, are encoded by different genes and function as transcription factors upon binding to the ligand oestrogen to activate or inhibit the expression of target genes [24].

Classical nuclear oestrogen receptors are widely expressed in various cell types such as hepatocytes, osteoblasts, cardiomyocytes, neurons, endothelial cells of blood vessels, and adipose tissue cells [25]. These receptors are activated by binding small lipophilic ligands such as steroid hormones, thyroid hormones, retinoids, and vitamin D3. Without a hormone, the receptor remains in the nucleus of target cells, bound to a heat shock protein complex that inhibits its activity. Ligand binding activates the receptor, forming stable oestrogen receptor dimers that interact with specific DNA regions called response elements located in the regulatory regions of target genes [26].

In addition to nuclear oestrogen receptors, new cell surface receptors such as GPR30 are involved in rapid cellular responses such as protein phosphorylation or kinase activation [27]. These rapid non-genomic signalling pathways may include activation of protein kinases such as mitogen-activated protein kinase (MAPK) and phosphatidylinositol 3-kinase (PI3K), leading to the activation of other transcription factors and changes in gene expression [28]. Mechanisms of oestrogen receptor action include interactions with other proteins such as coactivators, corepressors, and other transcription factors. For instance, ER-α and ER-β can heterodimerise to regulate the expression of target genes [29]. Oestrogens from the ovaries play significant roles in regulating the reproductive system primarily through interactions with nuclear oestrogen receptors. In contrast, brain-derived oestrogens protect against neuronal damage caused by harmful factors through both nuclear and cell surface receptors [30].

Oestrogen receptors α and β are structurally and homologously similar, with minor differences in their ligand-binding domains [31]. Both types of receptors are found in the circulatory, excretory, nervous, respiratory, and skeletal systems [32]. There is a significant predominance of β receptors in the hippocampus and gastrointestinal system. In contrast, α receptor expression is predominant in the uterus, ovaries, mammary gland, epididymis, testes, prostate, heart, aorta, liver, kidneys, and adrenal glands [28]. Transcriptional regulation by oestrogen receptors occurs through several mechanisms: the classical pathway, where the oestrogen receptor binds directly to oestrogen response elements (EREs) in gene promoters; the non-classical pathway involving the AP-1 protein, where the oestrogen receptor acts as a coactivator; and phosphorylation of serine and tyrosine residues on the receptor and activation of cytoplasmic kinase cascades [29]. Like other nuclear receptors, oestrogen receptors require transcriptional cofactors (coactivators and corepressors) such as Fos and Jun, which bind to the receptor and interact with other transcription factors [26]. In the classical genomic response, oestrogen binds to intracellular oestrogen receptors (ER-α and ER-β), which dimerise and translocate to the cell nucleus, regulating the transcription of target genes containing oestrogen response elements in their promoters. In this way, various signalling pathways are activated. These signalling pathways can also be initiated by binding oestrogen to receptors located on the cell membrane [33]. In addition to binding to nuclear receptors, oestrogens also act on membrane receptors located in the cell membrane of neurons, enabling a wide range of biological functions, including neuronal protection and regulation of reproductive functions, contributing to healthy ageing and diverse outcomes in oestrogen therapy for age-related diseases [25].

## 4. The Role of Oestrogen in the Physiology of Women and Men

Nuclear oestrogen receptors (ER-α and ER-β) and the GPER regulate both female and male reproductive organs and other non-reproductive tissues [34]. Through their receptors, oestrogen plays a crucial role in many physiological processes in both women and men [16]. ER-α is primarily located in the uterus and pituitary gland, regulating neuroendocrine and reproductive functions. In contrast, ER-β is mainly expressed in ovarian granulosa cells and plays a crucial role in ovarian function. A deficiency in ER-α leads to severe fertility issues in females, while the lack of ER-β results in infertility due to ovarian defects. Hamilton et al. noted that the diversity of oestrogen’s mechanisms of action is critical for their function in various biological contexts, which has significant clinical implications [35]. The ovaries are the primary source of circulating oestrogens in females. In contrast, in males, the testes produce only about 20% of circulating oestrogens, with the remainder coming from local production by adipose tissue, the brain, skin, and bones, where testosterone (T) is converted to oestrogens by the action of aromatase [36]. Aromatase is present in male reproductive cells and the cytoplasmic droplets of sperm tails, but its activity decreases as the sperm traverse the epididymis [37]. The oestrogens produced via aromatisation are deactivated by sulfoconjugation, catalysed by oestrogen sulfotransferase, present in the testes, epididymis, and vas deferens [38]. Oestrogen receptors play a crucial role in the response to oestrogens during the ontogenetic development of both females and males. Research indicates that male foetal reproductive organs are sensitive to oestrogen effects at an early stage of their development when these organs are still sexually undifferentiated [39]. As structures such as the prostate, bulbourethral gland, epididymis, vas deferens, and seminal vesicles differentiate themselves from one another, the expression of oestrogen receptors persists. Consequently, male foetal reproductive organs and their early forms are susceptible to the action of both endogenously produced and externally sourced oestrogens [40]. Table 2 compares the differences between women and men in oestrogen physiology.

## 5. Oestrogens and Bones

Oestrogens indirectly affect bone metabolism by influencing the production of interleukin 1 and 6 (Il1, Il6), TNFα, and granulocyte-macrophage colony-stimulating factors (GM-CSFs). They also stimulate the release of insulin-like growth factor 1 (IGF-1), transforming growth factor β (TGFβ), and procollagen-1 from osteoblasts. Oestrogen deficiency is responsible for the postmenopausal decline in bone mass and is associated with an increased frequency of bone tissue remodelling [41]. Studies have demonstrated the importance of oestrogen deficiency in the aetiology of bone loss in postmenopausal women. Albright [42] observed that oestrogen deficiency is the cause of postmenopausal osteoporosis. He proposed that menopause is not caused by the natural ageing process or the surgical removal of the ovaries, as previously thought, but results from the inhibition of ovarian function and a sudden decrease in oestrogen levels, which subsequently contributes to bone loss [43]. He also characterised the mechanism of osteoporosis pathogenesis and proposed an oestrogen supplementation strategy for its treatment. The hypothesis concerning the role of oestrogens in the pathogenesis of osteoporosis has been supported by subsequent studies, which showed that oestrogen administration prevents bone mass loss induced by ovary removal in premenopausal women [44]. It was also demonstrated that bone cells are a target site for oestrogen action [45]. Besides oestrogen deficiency, other factors contributing to age-related bone mass loss in women include secondary hyperparathyroidism [46]; impaired osteoblast function, likely caused by changes in local cytokine concentrations [47] or systemic growth factors [48]; and deficiencies in certain dietary nutrients, such as vitamin D and calcium [49].

Research is being conducted to explain how oestrogen deficiency contributes to the pathogenesis of secondary hyperparathyroidism and increased bone turnover, leading to slow bone tissue loss in postmenopausal women. Results indicate dependencies and interactions between oestrogen deficiency and bone tissue function. Prestwood et al. [50] demonstrated that short-term oestrogen administration to women (over 80 years old) significantly reduces the biochemical markers of bone turnover. McKane et al. [51] conducted a clinical study analysing the impact of oestrogens on bone tissue condition in women. They observed that parathormone (PTH) levels and bone resorption remained unchanged after correcting oestrogen deficiency. Similar effects were observed by Khosla et al. [52], showing that in postmenopausal women not receiving oestrogens, PTH levels and bone turnover markers increased sharply, unlike in women receiving oestrogen therapy. Oestrogens affect bone turnover directly by acting on bone cells and indirectly, probably by influencing extraskeletal calcium homeostasis [53]. Barger-Lux et al. [54] showed that calcium absorption correlates significantly with growth, body mass, and oestrogen levels. Numerous studies have examined the impact of oestrogen deficiency on extraskeletal calcium homeostasis. Gallagher et al. [55] observed that plasma total 1,25(OH)2D concentration and calcium absorption increased in postmenopausal women receiving oestrogens. Some researchers attribute the increase in plasma total 1,25(OH)2D levels after oral oestrogen administration to the increased amount of vitamin D-binding proteins induced by the so-called “first pass stimulation” of hepatic synthesis [56]. Others suggest that under oestrogen influence and the alteration of vitamin D metabolism, free 1,25(OH)2D plasma concentration increases [57]. Furthermore, Gennari et al. [58] demonstrated that in perimenopausal women before and six months after ovary removal, calcium absorption weakened due to oestrogen deficiency increasing in response to 1,25(OH)2D administration. The authors suggest that oestrogen directly affects the increase in intestinal calcium absorption. Nordin et al. [59] found that in postmenopausal women, “renal calcium leak” occurs due to oestrogen deficiency. In contrast, Cosman et al. [60] observed that oestrogens directly affect parathyroid hormone secretion. Oestrogen deficiency is now considered the primary cause of type I osteoporosis. Järvinen et al. [61] proposed another explanation for the role of oestrogens in controlling bone mass in women. The authors suggested that the initiation of oestrogen secretion during puberty mechanically “packs” excess bone-building components for reproductive processes (pregnancy and lactation). According to this hypothesis, during the reproductive period, the calcium deposit is “unpacked”. During menopause, an accelerated phase of bone tissue loss and the development of type I osteoporosis occurs [61].

## 6. Oestrogens and the Cardiovascular System

Oestrogen exerts pleiotropic effects on the cardiovascular system. Its action varies depending on the receptor type and location within the cardiovascular system. In the cardiovascular system, ER-α and ER-β are present in the plasma membrane, cytoplasm, and nucleus [62]. It has been confirmed that oestrogen prevents apoptosis and necrosis of cardiac and endothelial cells, reduces cardiac hypertrophy, and may have significant anti-inflammatory benefits in the ageing body [63].

Oestrogen stimulates the production of nitric oxide (NO), as demonstrated in endothelial cells of the aorta in mice with ER-α receptor knockout [64]. Through membrane oestrogen receptors, oestrogens activate signalling cascades crucial for cell survival and death, such as PI3K, Akt, and ERK 1/2 [65]. Interesting observations regarding the G protein-coupled oestrogen receptor (GPER) in vascular response to oestrogens have been made. Activation of this receptor by oestrogen induces NO production and vessel relaxation independently of the endothelium [66], thus protecting mitochondria and cardiac function [67].

The protective influence of oestrogens on the cardiovascular system is also associated with regulating lipid profiles by lowering LDL cholesterol levels and increasing HDL cholesterol [68]. Oestrogen prevents atherosclerosis by inhibiting smooth muscle cell proliferation through ER-α receptors and reducing monocyte adhesion to endothelial cells [69]. The positive effects of oestrogen on the heart are further evidenced by better survival rates in women with heart failure compared to men [70].

Molecular mechanisms responsible for oestrogen’s protective effects on cardiomyocytes involve regulating heat shock proteins (HSP), thereby protecting cardiomyocytes from apoptosis induced by stress and mitochondrial dysfunction [71]. Oestrogens inhibit matrix metalloproteinases (MMP) and enhance the action of MMP inhibitors (TIMP) [72].

## 7. Oestrogens and Brain

The oestrogen levels in the brain can be assessed in animals’ brains using radiological techniques to prepare brain images in people and postmortem in humans based on aromatase expression or via the evaluation of oestrogen receptor expression in brain tissues. As a result, 17-estradiol (E2) has been confirmed as a neurosteroid produced in neurons and astrocytes in the brain of many different species, as evidenced by in vivo studies in rats, songbirds, humans, frogs, and non-human primates [73]. 

The highest aromatase expression was found in humans using positron emission tomography (PET) imaging in the hypothalamus, amygdala, pons, midbrain, hippocampus, temporal cortex, and frontal cortex [74,75]. 

Oestrogens significantly influence cognitive and neural functions, impacting various aspects such as verbal activity, memory maintenance, spatial task performance, and motor skills. Moreover, they play a pivotal role in the pathogenesis of neurodegenerative disorders like Parkinson’s disease and dyskinesias, often associated with disruptions in oestrogen metabolism, particularly notable during the postmenopausal phase in women [76,77]. Additionally, oestrogens are implicated in depression, affecting both its occurrence and treatment [78,79]. Gender differences in depression prevalence emerge post-menarche and persist throughout the reproductive years, gradually declining after the perimenopausal phase. Hormonal fluctuations or oestrogen instability (e.g., premenstrually, postpartum, and perimenopausally) are linked to heightened susceptibility to depression in vulnerable women [80].

The localisation of oestrogen receptors ER-α and ER-β within the brain exhibits a diverse distribution pattern. ER-α is primarily concentrated in specific regions, such as the hypothalamus’s ventromedial nucleus (VMN) and the arcuate nucleus. In contrast, ER-β shows a broader distribution, encompassing areas like the hippocampus, neocortex, cerebellum, and various hypothalamic nuclei. This heterogeneous distribution underscores the multifaceted roles of oestrogen signalling pathways in regulating neural functions and behaviours across different brain regions. ER-α and ER-β are also co-expressed in regions including the preoptic area, bed nucleus of the stria terminalis, lower brainstem, and dorsal horn of the spinal cord [81]. 

ER-α and ER-β are strongly expressed during neurogenic development, especially in the developing brain. Studies in rats have shown ER is responsible for brain masculinisation, as most androgens produced by developing testes are converted to E2 by aromatase in rodents’ brains [82]. After birth, ER levels in the brain decrease and their expression is limited to various brain regions. In adult rodents, ER-α and ER-β generally show low expression levels but with a wide distribution pattern. In humans, masculinisation is mainly mediated by brain androgen receptors (AR), although the role of ER is not excluded. Significant species differences exist in the neuroendocrine control of adult brains, such as the localisation of aromatase and ER distribution in various brain regions in humans and rodents [82]. Oestradiol (E2) has been shown to influence synaptic transmission, dendritic spine formation, and long-term synaptic potentiation (LTP) through non-genomic and genomic actions. Studies in animal models have shown that the lack of ER-α leads to sexual behaviour disorders. In contrast, the lack of ER-β leads to spatial learning disorders and increased anxiety-like behaviour [83]. 

Ageing is associated with a decrease in ER-α and ER-β levels in synapses of hippocampal neurons in female rats. A decrease in ER activity in the brain may increase susceptibility to adverse neurological events later in life, especially in menopause, and a reduction in oestrogen levels [84]. There are apparent gender differences in the risk and progression of neurodegenerative diseases such as Alzheimer’s disease (AD) and Parkinson’s Disease (PD). Oestrogen shows neuroprotective effects in experimental models of AD and PD through the activation of different ER subtypes, including ER-α and ER-β [85]. In vitro studies have shown that oestrogens exert protective effects on neuronal cells, possibly resulting from their action against free radicals. In animals that do not have ER-α, oestrogens had no neuroprotective effects, suggesting that this oestrogen receptor is essential for the transduction of positive oestrogen effects observed in cells [76]. In humans, the beneficial effect of oestrogens on cognitive functions may be associated with neuronal protection in the cortex against apoptosis or improvement in intracellular signalling mechanisms, such as the MAP kinase pathway [77]. However, the exact neuroprotective mechanism of oestrogens in humans is not fully understood [77]. 

There are significant gender differences in the ageing process of the brain, where the age-related loss of brain tissue in the hippocampus and occipital lobes is more significant in women than in men [79,86]. Women also have more significant declines in hippocampal glucose metabolism with age, which is crucial for cognitive function and may be associated with AD, which has a higher prevalence in women [86]. Studies suggest that oestrogens may protect against cognitive decline through various mechanisms, such as increasing dendritic spine density, influencing cholinergic activity in the brain, and protecting neurons from damage caused by oxidative stress or excitotoxic stimuli [76,77,86]. However, studies on hormone replacement therapy (HRT) have yielded mixed results, and the benefits may depend on the age of initiation, the type of oestrogens used, and the presence of menopausal symptoms [76,77,86]. Studies on oestrogens have shown that they have a significant impact on cognitive and neurological functions in women. 

Neuroendocrine studies suggest that oestrogens may affect the central functioning of serotoninergic and cholinergic systems [81]. Structural and functional brain imaging techniques have confirmed that in healthy women, oestrogens can modulate cerebral blood flow, glucose metabolism, and neurotransmitter systems [81]. A decrease in oestrogen levels in the blood with age, especially after menopause, often accelerates ageing effects on cognitive functions such as memory, concentration, and information processing speed [79,86]. 

Oestrogens affect various aspects of brain function, including memory and motor skills [76]. Sherwin et al. noted that oestrogens have a beneficial effect on memory in women [77]. Oestrogens also play a role in regulating neurotransmitters [81]. Smith et al. pointed out the influence of oestrogens and progesterone on GABA and glutamine reactions in various brain areas [81]. In addition, studies by Klaiber et al. suggest that serum oestradiol levels are associated with mood and menopausal time during hormone therapy [78]. The cognitive effects of oestrogen therapy have been the subject of numerous studies. For example, Schneider et al. examined the effect of oestrogen replacement therapy on the response to fluoxetine in depression in older people, which showed positive results [79]. Reiman et al. used positron emission tomography to study the normal menstrual cycle, which also confirmed the influence of oestrogens on the brain [86]. Furthermore, the oestrogenic influence on ageing processes and neurodegenerative diseases such as AD is well documented. Yaffe et al. showed that oestrogens may be associated with reducing the risk of cognitive decline in women [87]. Other studies, such as those conducted by Asthana et al., demonstrated the positive effects of transdermal oestrogen replacement therapy in women with AD [88].

## 8. Oestrogen and Immunity

The immune system exhibits sex differences in the intensity of immune responses to external antigens and in the progression of autoimmune reactions [89]. Oestrogen influences the modulation of innate immunity by regulating the number and function of immune cells such as neutrophils and macrophages and the induction of cytokines such as TNF-α, interleukins, T and B cells, including CD4+ (Th1, Th2, Th17, Treg) and CD8+, as well as interferonγ (IFNγ) [90]. The impact of oestrogens on autoimmune diseases varies depending on the disease context. In multiple sclerosis (MS), oestrogen exhibits anti-inflammatory and neuroprotective effects, whereas in systemic lupus erythematosus (SLE), oestrogen can exacerbate the disease [91]. Oestrogen shows a protective effect in multiple sclerosis by reducing pro-inflammatory cytokines and inhibiting inflammation and demyelination. It also alters gene expression in spinal cord tissue [92]. Non-organ-specific autoimmune disease (SLE) acts oppositely, increasing disease severity, relapses, antibody production, and inflammation [93].

Oestrogens influence immune responses against viral infections through the regulation of various mechanisms. Among others, in the early stages of the immune response, they impact the negative regulation of the NF-κB signalling pathway, inhibition of type I interferon (IFN) release, and direct suppression of IFN signalling through the activation of GPER1 receptors [94]. Additionally, they can regulate the levels of pro-inflammatory cytokines by blocking NF-κB signalling [95].

It has been observed that women, due to the protective effects of oestrogens, are less susceptible to developing severe sepsis following an infection [96]. In vitro studies have confirmed the homeostatic function of oestrogens in inflammation development, involving activating some responses while enhancing others [97]. In vivo, in mice with malaria, oestrogen administration elevated IFN-γ levels, reducing parasitaemia [98]. Clinical observations in humans confirm a lower susceptibility to infections, including parasitic infections, in women than in men [99]. Furthermore, oestrogens can influence macrophage profiles, accelerating wound healing processes and recovery [100]. Oestrogens and their receptors regulate adaptive immune responses to infections and adaptive immunity by raising circulating antibody levels [101]. Immunohistochemical studies have confirmed that B and T lymphocytes exhibit high ER expression [102].

## 9. Exogenous Sources of Endocrinologically Active Oestrogen Derivates Compounds

In the external environment, besides exogenous oestrogens found in animal products, some chemicals disrupt endocrine functions (EDCs) [103]. EDCs are exogenous substances that affect the functioning of organisms [104]. These substances are widely used in industry and disrupt biological systems’ homeostasis upon entering the environment [104]. They possess a phenol group, which makes them similar to steroid hormones and allows them to interact with hormone receptors [105]. Oestrogen-mimicking EDCs are called xenoestrogens because they interact with ER-α, ER-β, and GPER. They are characterised by good solubility in fats, facilitating their accumulation in the body [103].

EDCs can induce a variety of changes at the molecular level, including chromosomal aberrations, DNA damage, epigenetic modifications, and influence on signalling pathways [106]. Oestrogen and other oestrogen receptor agonists can bind to them in a mechanism directly binding to a specific sequence called an ERE [107].

Oestrogen plays a pleiotropic role in regulating numerous signalling pathways and modifying the expression of multiple genes. Variants in these genes can be predisposed to overexpression or be suppressed through the involvement of various epigenetic and enzymatic mechanisms [108]. The search continues for new gene variants or their polymorphisms that could serve as markers of damage or the impact of EDC exposure on the organism. Monitoring such markers will help assess the risk of developing endocrine disorders and oestrogen-dependent diseases.

### 9.1. Bisphenols

The most commonly studied oestrogen-mimicking compounds include bisphenol A (BPA), 4-para-nonylphenol (NP), polychlorinated biphenyls (PCBs), organochlorine pesticides, bisphenol S (BPS), diethylstilbestrol, dichlorodiphenyltrichloroethane (DDT), and di(2-ethylhexyl) phthalate (DHEP). Bisphenols significantly impact numerous pathways and genes responsible for proliferation, the cell cycle, apoptosis, and transcription in the body [109]. 

#### 9.1.1. The Influence of Bisphenols on Cancer Development 

BPA’s ability to mimic oestrogen and disrupt normal hormonal signalling in breast tissue highlights its potential role as a significant environmental risk factor for breast cancer. The correlation between bisphenol concentrations and breast cancer incidence remains unexplained [110]. 

BPA has been identified as a significant risk factor for breast cancer, known for its oestrogen-mimicking properties, it interacts with ERs in breast tissue, thereby influencing critical signalling pathways involved in mammary gland development and function. Studies have shown that BPA exposure increases oestrogen sensitivity and upregulates ER expression, leading to inappropriate activation of oestrogen signalling pathways implicated in breast cancer development.

Moreover, BPA has been linked to enhanced proliferation, migration, and invasion of breast cancer cells. It activates pathways such as ERK1/2 and AKT, which promote cell growth and survival and induce the expression of MMPs that facilitate tumour cell invasion and metastasis. These effects underscore BPA’s role in fostering a microenvironment conducive to cancer progression [111,112].

In addition to its direct effects on cellular processes, BPA exposure has been associated with epigenetic modifications, including aberrant DNA methylation. Such alterations can affect the expression of genes like *LAMP3*, which are known to play crucial roles in breast cancer pathology. Furthermore, BPA influences the expression of genes involved in cell cycle regulation, apoptosis, and hormone receptor activities, including *HOXB9*, implicated in promoting cell proliferation and neovascularisation [113].

Beyond its effects on gene expression and cellular processes, BPA alters stem cell differentiation and affects the breast microenvironment by modulating signalling pathways critical for mammary fibroblast development and altering the expression of receptors involved in growth factor signalling. These changes collectively contribute to an increased risk of breast cancer development and progression [112].

It is particularly challenging to comprehensively depict all genetic variants, their polymorphisms, mutations, and epigenetic changes due to their substantial quantity. Therefore, selected molecular targets are evaluated using various molecular techniques, including the assessment of expression levels, methylation, epigenetic silencing by small miRNAs, evaluation of restriction fragment length polymorphism (RFLP), sequencing and microarrays. An illustrative list of genes and their corresponding single nucleotide polymorphisms (SNPs), identified via oestrogen metabolism, has been analysed for their association with mammographic density in postmenopausal women. Among them, candidate genes potentially acting as markers for cancer development due to their activation by exogenous oestrogens were identified. The analysis included *CYP11A1* (rs11638442, rs16968478, rs2279357, rs2959003, and rs2959008), HSD17B3 (rs2066485 and rs7039978), *NQO1* (rs1469908), and *STS* (Steroid Sulfatase) (rs17268974, rs2270112, and rs707762). Another significant gene was *CYP1A1*, which plays a crucial role in oestrogen metabolism. The results of the analysis indicated that some of these SNPs showed significant associations with breast density, suggesting their potential role as risk markers or factors influencing breast cancer development [111].

Using SAGE or microarray techniques [112], scientists have identified a vast number of ER target genes and revealed that a significant portion of these genes are silenced following induction by oestrogen-like compounds or oestrogen itself. For instance, genes involved in cell cycle pathways, pro-apoptotic actions, as well as cytokines and growth factors that inhibit proliferation, are downregulated upon oestrogen induction [113].

Moreover, the activation of oestrogen receptors by oestrogen-like ligands can contribute to the activation of ERE sequences in genes that are characteristic of carcinogenic processes [114]. Proteins encoded by these genes include oestrogen-responsive finger protein (Efp), cytochrome c oxidase subunit VIIa-related polypeptide (*COX7RP*) and oestrogen receptor-binding fragment-associated antigen 9 (*EBAG9*). Binding to oestrogen receptors is also responsible for the expression of *FOXA1* and *FOXP1*, which promote the proliferation and migration of breast cancer cells. Other targets activated through ERE include transcription factors *JUN*, *FOS*, *PGR*, and *TP53*; intracellular signalling molecules HRAS, BCL2, and BRCA1; enzymes CHAT, NQO1, and CKB; secreted proteins LTF, SCGB1A1, OVGP1, C3, and AGT; hormones LHB, OXT, PRL, and AVP; and membrane proteins SNAT2, VEGFA, mitogen TFF1, and protease CTSD [114,115]. Bisphenols affect various cancerogenesis processes through oestrogen-regulated and independent pathways (MAPK, STAT3, and PI3K/AKT) [106].

Other bisphenols, bisphenol B, Z, and 4 MeBPA, also bind to the same active site of the oestrogen receptor alpha as oestradiol but show lower affinity than the natural hormone [116]. 4MeBPA and BPA showed weaker receptor activation, while BPZ inhibited its activation [117]. Consequently, they affect the increased expression of 14 genes (*CEL SR2*, *FOSL 2*, *JUN*, *HSPA 13*, *IER3*, *ADORA1*, *DDIT4*, *IGF1R*, *PGR*, *RUNX2*, *SLC7A11*, *SLC7A2*, *SLC7A5*, and *STC2*) and decreased expression of three genes (*BCAS3*, *PHF19*, and *PRKCD*) [118]. The impact on these genes and their significance is summarised in (Table 3). BPA has been shown to cause excessive migration and proliferation of MCF-7 cells by influencing *PTTG1* and *CDC20* genes, which guard the proper cell cycle [119]. Additionally, the mechanism leading to decreased expression of miR-381-3p, resulting in increased activity as seen in breast cancer cells, has been discovered [119].

#### 9.1.2. The Influence of Bisphenols on Endocrine Disorders 

BPA and its derivatives are used in the production of plastics such as dental materials, plastic bottles, receipts, and food packaging [115]. Although the use of BPA in specific products has been banned in many countries, substitutes like bisphenol B or tetramethyl bisphenol A (4 MeBPA) also exhibit endocrine activity [116]. BPA exerts its effects by disrupting hormonal functions through various pathways, primarily via thyroid hormone and glucocorticoid receptors. These interactions can lead to dysregulation of hormone levels, changes in gene expression, and potential long-term effects on metabolic and reproductive health. Further research is needed to fully elucidate the mechanisms and consequences of BPA exposure, particularly in the context of human health and development [115,116].

BPA has been shown to disrupt hormonal pathways by interacting with thyroid hormone (TH) receptors and glucocorticoid receptors (GR), which has significant implications for endocrine health. BPA acts as an antagonistic ligand for thyroid hormone receptors, competitively binding to them and inhibiting transcriptional activity stimulated by triiodothyronine (T3). In vitro studies have demonstrated that BPA can also suppress thyroid hormone receptor-mediated gene expression by enhancing the recruitment of the corepressor N-CoR to the thyroid hormone receptor, even at low concentrations of this compound. Offspring of rat mothers exposed to BPA during gestation and lactation showed increased thyroxine (T4) levels and expression of thyroid hormone-responsive genes in the brain. A loss of negative feedback via one thyroid hormone receptor isoform was hypothesised to be responsible for the increase in T4 levels [116]. Additionally, in vivo studies have demonstrated elevated thyroid hormone levels in male offspring of rat dams exposed to BPA through drinking water. Halogenated derivatives of BPA, such as tetrabromobisphenol A and tetrachlorobisphenol A, used as flame retardants, have also been shown to inhibit T3 binding to thyroid hormone receptors, acting both as agonists and antagonists of these receptors. BPA can also interact with glucocorticoid receptors, acting as an agonist for GR similar to dexamethasone and cortisol. In vitro studies have shown that BPA stimulates adipogenesis in 3T3-L1 cells through the activation of the glucocorticoid receptor. There is also evidence of increased corticosterone levels in female offspring of rat mothers exposed to BPA during pregnancy and lactation, potentially affecting the hypothalamic–pituitary–adrenal axis function [120].

BPA also exhibits antiandrogenic activity by binding to androgen receptors and binds to the aryl hydrocarbon receptor (AhR), which affects xenobiotic metabolism as well as steroid synthesis and metabolism. Additionally, BPA inhibits aromatase activity, which could decrease the conversion of testosterone to oestradiol and may have implications during development and adulthood [116].

#### 9.1.3. The Influence of Bisphenols on the Male Reproduction System 

BPA disrupted meiotic processes, leading to chromosomal aberrations, impaired chromosome synapsis, and disrupted double-stranded DNA break repair mechanisms during cell division, resulting in aneuploidy [120]. Another bisphenol, BPAF, activates the oestrogen receptor alpha more strongly than BPA and acts as an androgen receptor antagonist [121]. Exposure to BPAF nine days after birth caused changes in the male reproductive system of mice through the activation of the oestrogen receptor [122]. BPAF disrupts the blood–testis barrier, initiates DNA damage, and induces multinucleation [123]. It can trigger autophagy and apoptosis and increase oxidative stress. Additionally, it reduces the number of Leydig cells and impairs their regeneration, affecting testosterone, LH, FSH, progesterone, and oestradiol levels in sera [124]. Another study showed increased oestradiol concentration, which may disrupt the hypothalamic–pituitary–gonadal axis via negative feedback [125]. BPAF increases the expression of kiss1, an activator of this axis and activates aromatase [126]. Moreover, BPAF and BPF caused changes in the cortex structure of the testes to resemble the ovarian cortex and testicular asymmetry in chicken embryos [127]. Many bisphenols, such as BPAF, BPBP, BPB, BPG, BPC, and BADGE, affect spermatogenesis by activating CatSper Ca^2+^, which is responsible for sperm fertilising ability [128].

### 9.2. Phthalates

Phthalates are compounds often used as plasticisers [129]. They are used to produce various pharmacologically active compounds to improve their bioavailability. Although recognised as toxic, phthalates, such as dibutyl phthalate (DBP), are added to medicines and dietary supplements [130]. Phthalates are not chemically bound and remain non-covalently attached to the polymer chain. Therefore, they can easily detach and migrate into food, air, or water [131]. In a study of urine samples from people taking phthalate-containing medications, the concentration of DBP was significantly higher than in those not using such medications [132]. They have been proven to interfere with endocrine functions. Certain phthalate metabolites, such as monobenzyl phthalate (MBzP) and mono-2-isobutyl phthalate (MiBP), may hypothetically reduce the risk of breast cancer by stimulating the PPAR receptor activated by gamma and alpha peroxisome proliferators in adipose tissue, which is responsible for its differentiation, storage, and cancer inhibition [133]. Furthermore, studies suggest that the presence of phthalates is potentially linked to increased expression of *ADAM33* gene through enhanced methylation of intron 1, also having anti-oncogenic effects [134]. Nevertheless, like bisphenols, they bind to the oestrogen receptor alpha and affect the cell cycle, mainly during the G2/M phase, similarly to bisphenols. An exception is benzyl cyclohexyl phthalate (BCP), which, unlike benzyl butyl phthalate (BBP) and butyl octyl phthalate (BOP), did not affect the cell cycle [117]. As a result of BOP’s interaction with the receptor in the MCF-7 cell line, the expression of eight genes (*CEACAM5*, *CYP1A1*, *DDIT4*, *IER3*, *KLHL24*, *SLC7A5*, *SLC7A11*, and *STC2*) was intensified, while the expression of seven genes (*ADORA1*, *CCNA2*, *CDK1*, *FKBP4*, *PGR*, *SFPQ*, and *TFAP2C*) was reduced [135] (Table 3). Additionally, it has been shown that BBP causes modifications in the SIPR3 protein and stimulates SPHK1, contributing to the formation of breast cancer stem cells and promoting metastasis [136].

### 9.3. Other EDCs

Organic phosphoric acid (OPEs) esters are often used as plasticisers and flame retardants [137,138]. Aromatic esters show significantly greater activity than those without benzene in their structure. Among them, two compounds, tri-o-cresyl phosphate (TOCP) and the less-active tri-m-tolyl phosphate (TMCP), showed a high likelihood of binding to the oestrogen receptor alpha. TOCP binds more strongly to the receptor (lowest binding energy: −10.05 kcal mol^−1^) than 17-β-oestradiol (lowest binding energy: −9.48 kcal mol^−1^). It potentially shows a higher affinity for the oestrogen receptor than its natural ligand [117]. Triphenyl phosphate (TPHP) and resorcinol bis(diphenyl) phosphate (RDP) cause proliferation in the MCF-7 cell line. TOCP also proliferated the MCF-7 cell line and affected the expression of 22 genes [118] (Table 4 and Table 5). The examples discussed above exhibited agonistic effects on oestrogen. Other esters and their metabolites: TCP, TPP, TiPP, DBP, DBzP, DPHP, DPPC, DPP, DBBP, MDCP, DICP, DECP, and DEAP, when administered together with 17-β-oestradiol, significantly inhibited the expression of *TFF1* and *EGR3* genes and did not cause proliferation in the MCF-7 cell line, indicating their antagonistic action towards ER [139].
ijms-25-08167-t003_Table 3Table 3Changes in the expression of selected gene variants during the induction and metastasis of carcinogenesis under the influence of endocrine-disrupting compounds with similar mechanisms of action to oestradiol-bisphenols BPA, BPB, BPZ, and 4MeBPa in MCF7 cell lines.GeneRegulation: Up/Down ↑/↓Cancerogenesis Progression:*ADORA1*↑excessive proliferation [140]*CELSR2*↑migration, invasion, adhesion, metastasis [141]*DDIT4*↑suppression of mTOR activity,excessive proliferation, apoptosis disorders [134,142]*FOSL2**AP-1*↑participates in activating the metastasis cascade [143]*HSPA13*↑TANK stabilisation, proliferation, migration [144]*IER3*↑apoptosis disruption [145]*IGF1R*↑inhibition of apoptosis, proliferation, enhanced ER activation [146,147]*JUN**AP-1*↑invasion, migration, metastasis [143]*PGR*↑abnormal and excessive cell growth [148]*RUNX2*↑bone metastasis, impaired function and development of osteoblasts [149]*SLC7A2*↑inflammation and oxidative stress caused by increased NO synthesis, but inhibited invasion and migration [150,151]*SLC7A5*↑provides the tumour with access to amino acid development and proliferation,activates the mTORC1 pathway [152]*SLC7A11*↑protection of cancer cells from oxygen radicals,delivery of glucose and glutamine to cancer cells [153]*STC2*↑increased proliferation and viabilityregulation of the MAPK pathway [154]*BCAS3*↓overexpressed gene in breast cancer (inhibits apoptosis, enhances proliferation) gets silenced [155]*PHF19*↓silencing of proliferation and migration,increased apoptosis [156,157]*PRKCD*(suppressor gene)↓accelerated development of cancer cells [158]

### 9.4. Phytoestrogens

The structure of phytoestrogens resembles that of oestradiol, allowing them to bind to oestrogen receptors [175]. Phytoestrogens can act as agonists or antagonists; hence, they are oestrogen receptor regulators. Unlike E2, most compounds in the phytoestrogen group exhibit greater affinity for the ER-β than for the ER-α [176,177,178]. In a study conducted by Jiang et al., it was also observed that the potency of phytoestrogens depends on the level of oestrogen receptor expression in the cell and indirectly on the concentration of endogenous oestrogens. Additionally, it was shown that gene expression modification at low doses of the phytoestrogens studied was primarily due to binding to the ER-β [178].

In contrast, at higher concentrations, no such preference was observed. The varying expression of ER-α and ER-β in different body cells determines their action [178]. The main groups of phytoestrogens present in our diet are isoflavones, prenylflavonoids, coumestans, and lignans found in food products (Table 6) [7].

#### 9.4.1. 8-Prenylonaringenin

8-Prenylnaringenin (8-PN) appears to be the phytoestrogen with the strongest oestrogen-like properties. The highest amounts of 8-PN have been found in the common hop and its derivative products, including beer. It is distinguished by a greater selectivity for the ER-α [179]. The biological effects of 8-PN may result from its oestrogenic properties or interactions with other mechanisms independent of ERs [180]. 8-PN has the potential to alleviate menopause-related symptoms and could be an alternative to hormone replacement therapy due to its lower risk of side effects. Additionally, 8-PN has shown the ability to inhibit platelet aggregation, suggesting its potential antiplatelet activity [181]. Prenylated naringenin has been identified as a substance capable of inhibiting steroidogenesis in Leydig cells, as demonstrated in in vitro studies conducted on mouse Leydig cells [182]. The effects of 8-PN are illustrated in Figure 1. 

#### 9.4.2. Genistein

Genistein, which belongs to the isoflavone group, exhibits estrogenic activity at least ten times lower than that of oestradiol [179]. In a prospective study on the impact of soy isoflavone intake on the risk of developing prostate cancer, it was shown that men who consumed the highest amounts of genistein (≥32.8 mg/day) had a lower risk of developing prostate cancer compared to men with a lower intake of this compound [183]. Additionally, a study conducted by Boutas et al. demonstrated a reduced incidence of breast cancer in women consuming high doses of soy isoflavones (>15 mg/day) compared to women consuming low amounts (0–15 mg/day). This study included both premenopausal and postmenopausal women [184].

The issue of the oestrogenic effect after consuming plants containing these compounds appears to be complicated due to differences in phytoestrogen content and difficulties in achieving sufficiently high concentrations in consumed products, the presence of other biologically active compounds in the plant material, and the lack of sufficient studies in this field [177].

#### 9.4.3. DHEA 

Peripheral synthesis of androgens and oestrogen in postmenopausal women is insufficient to raise their blood levels due to enzymatic inactivation. The precursor for intracellular synthesis of oestrogens and androgens is DHEA, which circulates primarily in its inactive phosphorylated form, DHEAS. This process does not lead to increased oestrogen levels in the blood, as only the degradation products of this hormone are released into the circulation. This mechanism helps prevent the stimulation of endometrial growth, which could increase the risk of endometrial cancer. Currently, no research confirms that DHEA is non-carcinogenic, which is why this treatment method is not recommended for hormone-related diseases [185,186].

The endogenous level of DHEA stabilises at a certain level. Low concentrations of DHEA exacerbate menopausal symptoms. Increasing the amount of DHEA in the body can only be achieved through external supplementation. In postmenopausal women, oestrogen levels in the blood must be kept low to reduce the risk of endometrial cancer; however, oestrogen must still be present in peripheral tissues. DHEA ensures the local synthesis of oestrogens in peripheral tissues. Approximately 95% of androgens and oestrogens produced locally are inactivated and degraded before being released into the bloodstream [186].

Oral or parenteral administration of DHEA leads to its conversion to DHEAS in the liver, rapidly increasing DHEAS levels. Conversely, oral administration of DHEAS has little impact on blood levels, as it undergoes hydrolysis in the stomach [185]. DHEA can convert to oestrogen in cells with oestrogen or androgen receptors. DHEA also has 5–10 times greater affinity for the ER-β than Er-α [186]. 

The continued metabolism of oestrogen in cells can lead to the formation of products such as 2-hydroxy oestradiol (2-OHE2) and 4-hydroxy oestradiol (4-OHE2) (Figure 2). Studies show that the latter form is more carcinogenic, generating reactive oxygen species and causing nuclear DNA damage by forming DNA adducts. Both 2-OHE2 and 4-OHE2 stimulate cancer cell growth. Imbalances in enzymes that metabolise oestrogen lead to increased synthesis of 4-OHE2. The key enzyme in this transformation is CYP1B1, whose expression is increased in breast cancer cells, promoting their proliferation by enhancing survival chances [187].

DHEA exhibits a wide range of potential physiological benefits. DHEA may improve endothelial function, enhance cellular immunity, reduce inflammation, and exert neuroprotective effects. Additionally, it shows protective effects on the cardiovascular system [188]. Although study results are conflicting, some suggest significant reductions in atherosclerotic plaque size and protection against pulmonary hypertension and heart remodelling. In the context of glucose metabolism, DHEA may have protective effects against glucose toxicity, suggesting its potential role in diabetes prevention. Furthermore, some studies indicate possible benefits in fat mass reduction and muscle strength improvement, although not all studies confirm these effects [188].

Some studies have shown that DHEA can reduce depressive symptoms in HIV patients [189] and positively impact middle-aged patients with both primary and mild depression [190]. Higher serum levels of DHEA were also associated with the prevention of type 2 diabetes in older men but not in older women [191]. DHEA’s potential efficacy is also in treating steroid-induced osteoporosis [192].

However, there is no convincing evidence for the benefits of DHEA in preventing cancer and atherosclerosis, treating obesity and diabetes, or its anti-glucocorticoid action [193,194]. Moreover, it does not bring significant benefits in increasing muscle mass in postmenopausal women nor does it improve cognitive abilities in HIV patients [195]. No benefits were observed in a study involving 60 women with inactive systemic lupus erythematosus (SLE) [196].

Additionally, some studies have shown no changes in lipid profiles following DHEA supplementation and even an increase in foam cell formation from macrophages, which may exacerbate atherosclerosis. Traish et al. emphasise the need for large, long-term, placebo-controlled clinical trials to robustly assess the safety and efficacy of DHEA in treating various conditions [188].

## 10. Oestrogen-Related Diseases

Excess oestrogen can lead to the development of various diseases, including breast, ovarian, endometrial, and prostate cancers, autoimmune diseases such as systemic lupus erythematosus and multiple sclerosis; as well as thromboembolic disease. Additionally, in men, it can cause hypogonadism, oligospermia, and gynecomastia [197,198,199].

### 10.1. Cancerogenesis

Oestrogens, by binding to ER-α and ER-β, which belong to the group of transcriptional activators, stimulate cell proliferation. Additionally, their metabolites, catechol oestrogens, can covalently bind to DNA. The binding of 2- or 4-hydroxy oestradiol to DNA can lead to the formation of apurinic sites and destabilisation of glycosidic bonds. Furthermore, the aromatic hydroxylation of oestrogens to their catechol derivatives is a source of reactive oxygen species [200].

#### 10.1.1. Breast Cancer

The active oestrogen-ER complex regulates various genes that play crucial roles in cell proliferation and cell cycle progression. The mitogenic effects of oestrogens are particularly evident during the transition from G1 to S phase, during which critical effectors of oestrogen action are c-Myc and cyclin D1 [201,202]. Inhibition of c-Myc or cyclin D1 activity eliminates oestrogen-stimulated proliferation of breast cancer cells, while induction of c-Myc or cyclin D1 can mimic the effects of oestrogens. Oestrogen inhibits apoptosis by upregulating the expression of anti-apoptotic Bcl-2 and Bcl-XL in breast cancer cells [203]. The oestrogen-ER-α complex interacts with proteins such as c-Src through non-genomic action and activates MAPK and PI3K/Akt pathways, which are classically associated with cell survival [204]. The rapid growth and division of cancer cells require adequate nutrient supply from the blood through angiogenesis. In breast cancer cells, angiogenesis is induced by interleukin 8 (IL-8) and vascular endothelial growth factor (VEGF), the secretion of which is stimulated by oestradiol [205,206].

#### 10.1.2. Endometrial Cancer

Cytochrome P450 plays a crucial role in the hydroxylation of oestrogens, leading to the formation of metabolites such as 2-, 4- and 16-hydroxyestrogens [197]. The enzyme CYP1B1 is responsible for converting oestrogens into 4-hydroxyestrogen, which exhibits carcinogenic potential, mainly due to its ability to induce DNA damage by creating apurinic sites [197]. Immunohistochemical studies have shown increased expression of CYP1B1 in endometrial cancer cells. Additionally, it has been found that women with endometrial cancer have higher oestrogen levels in circulation compared to healthy women [200]. Factors contributing to the dysregulation of oestrogen levels include obesity, polycystic ovary syndrome (PCOS), oestrogen therapy without progestogens and diabetes, which may influence the risk of developing such tumours [207].

### 10.2. Venous Thromboembolism 

Venous thromboembolism (VTE) constitutes one of the most significant complications associated with the use of oral contraceptives and hormone replacement therapy [208]. The predominant symptom of this condition is deep vein thrombosis (DVT), which can lead to pulmonary embolism (PE). Initially, it was speculated that oestrogens might reduce the risk of cardiovascular diseases due to their beneficial effects on lipid metabolism [209]. However, the cardiovascular effects of oestrogens extend beyond this role. Oestrogens also affect blood vessel walls, stimulating the synthesis of nitric oxide and prostacyclin and inhibiting the action of calcium channels [210]. Studies have shown that therapy using conjugated oestrogens is associated with a lower risk of coronary heart disease compared to treatment combining oestrogen with progestogen [211]. Additionally, the use of oral contraceptives leads to an increase in the concentration of factor VII and IX but does not affect the concentration of factor VIII. The impact of oral contraceptives on protein S and protein C levels remains inconclusive. Decreased levels of these proteins are often observed, although reports indicate no changes or even an increase in their concentrations [212].

### 10.3. Autoimmune Diseases

Observations suggest that women exhibit a more robust immune response and are more susceptible to the development of autoimmune diseases compared to men. It is suggested that female sex hormones play a significant role in this increased immunological reactivity. Hormones such as oestrogens influence both immature immune cells (through interaction with the thymus and bone marrow) and mature cells, modifying the function and phenotype of T and B cells, immunoglobulin levels, and potentially the kinetics of immunoglobulin or cytokine production. Oestrogens may also affect markers of immune activation, such as myeloperoxidase (MPO), and pro-inflammatory cytokines, such as Il-6, TNF-α, Il-1β, IFN-γ, and GM-CSF. It has been shown that the induction of MPO release from inactive cells by E1, E2, and E3 stimulates the production of oxidants, even in the absence of pathogens [213].

Oestrogen hormones exhibit both immunostimulatory and immunosuppressive properties [214]. Systemic lupus erythematosus (SLE) has been observed predominantly in women of reproductive age [215]. Pregnancy, however, has been associated with disease exacerbations [214]. Costenbader et al. [216] observed that reproductive factors linked to increased SLE risk included early age at menarche, oral contraceptive use, early menopause, surgical menopause, and postmenopausal hormone replacement therapy [214]. High oestrogen levels in SLE patients led to increased autoantibody production and disease activation [217]. Autoantibodies and rheumatic symptoms were also more common in women using high-dose synthetic-oestrogen oral contraceptives [216].

The role of oestrogen and its metabolites in the pathogenesis of rheumatoid arthritis (RA) is debated, as oestrogen hormones have both suppressing and stimulating effects on chronic joint inflammation [218]. Experimental models of rheumatoid arthritis have shown beneficial effects of oestrogen [219]. In clinical studies, oestrogen did not activate existing RA, and postmenopausal hormone therapy improved the health status of RA patients and reduced osteoporosis [220].

## 11. Exogenous Oestrogen Therapies and the Risk of Breast Cancer

Oestrogen plays a significant role in the development of hormone-dependent tumours [221]. There is a recognised association between chronically elevated oestrogen levels in the body and the incidence of breast cancer [222]. It is suggested that hormone replacement therapy, oral contraception, in vitro fertilisation, and hormonal stimulation in transgender individuals may increase the risk of breast cancer [223,224,225].

In a cohort study of the Korean population, an increased risk of breast cancer was noted in women undergoing hormone replacement therapy (HRT) compared to those who had never received it. The risk coefficient increased with the duration of treatment. Among those receiving HRT, lower risk was observed in overweight women compared to women with average body weight [226]. A study in the British population also linked the use of HRT to a higher risk of breast cancer incidence, with the risk being higher in women using oestrogen–progestogen preparations than in those using oestrogen alone. The risk coefficient also increased with the duration of therapy. In the case of treatment with oestrogen–progestogen preparations for over five years, the highest risk was observed among women with a BMI < 25 and the lowest was observed in the BMI = 25–30 group [227].

In the case of hormonal contraception, specific dependencies have also been observed. An analysis of data from 1.8 million women in the Danish population calculated that among women who had used hormonal contraception before, the risk coefficient for developing breast cancer becomes statistically significant after at least five years of use. Additionally, among women who discontinued hormonal contraception after at least five years, the increased risk of breast cancer seemed to persist for another five years. Among combined preparations, the highest risk of occurrence was noted in those taking oestradiol and dienogest and, among progestin preparations, in those taking levonorgestrel [228].

Exogenous oestradiol is administered during in vitro fertilisation (IVF) preparations to create favourable conditions for embryo implantation in the uterus [224]. Studies have not found a significant increase in the risk of breast cancer among women who have had undergone IVF before [229], although a study by Stewart et al. observed that the risk coefficient was higher the younger the patients were at their first IVF attempt. An inverse correlation was observed by Katz et al., where the highest risk of breast cancer was associated with age over 30 at the time of the first IVF [230]. The van den Belt-Dusebout et al. study also considered the number of IVF cycles and response to the first cycle (number of retrieved oocytes). Patients who underwent seven or more cycles and those with a poor response to one cycle (less than four retrieved oocytes) showed a lower risk coefficient [231].

In pharmacological infertility treatment (ovulation induction), gonadotropins are used, which can cause an increase in oestrogen levels during the follicular phase. High oestrogen levels combined with high progesterone levels caused by simultaneous ovulation from multiple follicles seem to create conditions conducive to the development of breast tumours [232]. In a study of women who had used ovulation-inducing drugs before, no significant increase in the risk of breast cancer was observed, except for treatment lasting at least six months or a history of treatment with human menopausal gonadotropin (hMG) [233].

Components of hormonal therapy for transgender women (male sex assigned at birth, female gender identity) include antiandrogens in combination with oestrogens. In a cohort study from Amsterdam, it was observed that trans women have a lower incidence of breast cancer than cis women (female sex assigned at birth, female gender identity) but more frequently than cis men. Interestingly, among trans men (mainly receiving testosterone), breast tumours occurred more frequently than in cis men but less frequently than in cis women [225]. The main findings of the above studies are summarised in Table 7. 

## 12. Oestrogen and Cortisol 

The adrenal glands are crucial in synthesising steroid hormones, oestrogen, and cortisol. Oestrogen and cortisol are produced from the precursor 17-α-Hydroxyprogesterone, although their biosynthesis occurs through different metabolic pathways [234]. Understanding the steroidogenesis pathways in the adrenal glands that govern the synthesis of oestrogen and cortisol is crucial for understanding their mutual relationships. Biochemical analogies between oestrogen and cortisol suggest the possibility of their mutual regulation. In the steroidogenesis process leading to the formation of 11-Deoxycortisol, the direct precursor of cortisol, the enzyme 21-α-Hydroxylase acts on 17-α-Hydroxyprogesterone. This reaction is irreversible in the oestrogen synthesis pathway. There is no possibility of reversing this pathway; consequently, oestrogen synthesis from cortisol is impossible. Nonetheless, the mutual regulation of these two hormones is intriguing and requires further research to understand how they influence each other [234].

In the human placenta, the process of oestrogen synthesis has a different course compared to the adrenal glands. Oestrogen is synthesised from DHEA through aromatisation catalysed by aromatase. Cortisol plays a vital role in this synthesis pathway [235]. It has been shown that cortisol influences the stimulation of aromatase expression through the cAMP/Sp1 signalling pathway. As a result, an increase in cortisol levels translates into a local rise in oestrogen levels in the uterine myometrium. This mechanism is essential in inducing uterine myometrial contractions, which play a crucial role in childbirth [235].

Cortisol exhibits a circadian rhythm, peaking within 45 min of awakening in 90% of healthy adults, then gradually declining throughout the day and starting to rise at night [236]. The cortisol rhythm can differ between healthy and ill individuals. In the case of Seasonal Affective Disorder (SAD), the cortisol rhythm is delayed, as shown by Avery and colleagues [237]. SAD participants had a delayed cortisol minimum of two hours compared to the control group. Phase differences between cortisol and prolactin were also demonstrated by Koenigsberg and colleagues [238], who found an advance in the cortisol phase by one hour in people with depression. In other studies, changes in the cortisol phase did not vary in illness states. In contrast, different rhythms, such as immune factors, growth hormones, and prolactin, were advanced or delayed relative to mood [239]. Early studies on the relationship between cortisol rhythm and sleep showed more minor phase differences between cortisol nadir and sleep onset in depressed individuals compared to controls [240] (Table 8).

Oestrogen also exhibits a circadian rhythm, with a morning peak and several ultradian harmonics throughout the 24 h [241]. During the menstrual phase, the oestrogen peak occurs later in the morning. The oestrogen rhythm is relatively unaffected by the menstrual cycle, except for the acrophase during the menstrual phase (Table 8).

Studies on oestrogen circadian rhythms investigated cortisol and oestrogen rhythms in women with preterm labour. Results showed that the cortisol rhythm was phase-delayed in preterm labour compared to term labour, with no differences in the oestrogen phase [242]. Bao and colleagues [241] compared circadian rhythms of cortisol and oestrogen in women with and without depression, showing no correlation between the phases in depressed women in the late luteal phase, suggesting a phase decoupling between cortisol and oestrogen in these women. Table 8 summarises the key differences between cortisol and oestrogen, referring to their circadian rhythms, changes in health and disease states, and interactions with other hormones.

Significant interactions between oestrogen and cortisol are also observed. Cortisol-binding globulin (CBG) plays a crucial role in cortisol transport in the body. Ethinyl oestradiol (EE), an oestrogen derivative, induces nearly a six-hundred-fold increase in protein synthesis in the liver, including CBG, compared to natural oestradiol (E2) [246]. Oral contraceptives also exert a significant influence on CBG synthesis. When using ethinyl oestradiol (EE), cortisol levels are increased. Ethinyl oestradiol leads to increased CBG protein synthesis, which binds cortisol, resulting in a transient decrease in free cortisol, the active form in the body. This phenomenon disrupts the negative feedback loop, stimulating the pituitary gland to secrete adrenocorticotropic hormones (ACTHs). This hormone, in turn, stimulates the adrenal glands to produce additional cortisol. Thus, despite the increased levels of CBG and cortisol in the body caused by oral contraception, specific homeostasis is maintained [246].

In the case of breast cancer, the vast majority is considered hormone-dependent due to high oestrogen receptor expression. Recent studies suggest that cortisol may serve as a biomarker associated with the risk of breast cancer recurrence. It was found that patients with low initial cortisol levels more frequently experienced cancer recurrences. Additionally, an increase in cortisol levels was observed in patients without recurrences, while those with recurrences experienced a decrease in cortisol levels [247].

The enzyme 17,20-ligase acting on 17-α-Hydroxyprogesterone catalyses the formation of androstenedione. Subsequently, androstenedione is acted upon by 17-β-hydroxysteroid dehydrogenase to yield testosterone. Another enzyme in this pathway is aromatase, which ultimately leads to the formation of oestradiol. On the other hand, if 17-α-Hydroxyprogesterone is acted upon by 21-α-Hydroxylase, it leads to the formation of 11-Deoxycortisol, which is then converted to cortisol by 11-β-Hydroxylase (Figure 3) [212].

## 13. Discussion and Conclusions 

Oestrogen, one of the vital sex hormones, plays a crucial role in the functioning of the body, being essential for growth, development, and survival. It is a hormone of life, influencing sexual development, reproductive system function, bone, heart, and brain health. Its actions are diverse and encompass many biological processes, making it indispensable for women’s and men’s health and well-being [18,248]. However, oestrogen also has a dark side, leading to a significant scientific dilemma.

Disruption of oestrogen homeostasis significantly contributes to the development of numerous diseases and disorders, whether due to elevated or decreased levels of these hormones. Elevated oestrogen levels are associated with the development of various conditions such as breast cancer, prostate cancer, PCOS, ovarian cancer, endometriosis, gastric cancer, pituitary cancer, thyroid carcinoma, and adenomatous goitre [18,248]. High oestrogen levels can also lead to mental disorders such as schizophrenia and autoimmune diseases like SLE and MS. Moreover, increased oestrogen levels can be associated with hormonal problems in men, such as male hypogonadism and oligospermia, as well as gynecomastia and obesity [213].

On the other hand, reduced oestrogen levels also lead to severe health disorders. Low levels of these hormones can cause osteoporosis [249], joint pain (arthralgia), and neurodegenerative disorders such as AD and PD [250]. Oestrogen deficiency can also contribute to the development of diabetes and pregnancy complications such as eclampsia. Various medications, including antiandrogens, anticonvulsants, immunosuppressive drugs (including glucocorticosteroids and methotrexate) and herbicides, can further exacerbate issues related to low oestrogen levels [250].

All these data highlight the crucial role of oestrogen homeostasis in maintaining health and the severe consequences resulting from its disturbances, emphasising the importance of maintaining appropriate levels of these hormones in the body.

Oestrogens are involved in cell survival, proliferation, or differentiation by binding to two types of receptors, ER-α and ER-β, which translocate to the nucleus to regulate transcription, and GPER1 (GPR30), which rapidly signals independently of transcription [248]. It might seem that proper transcriptional and signalling regulation should establish oestrogen homeostasis. Meanwhile, studies show the existence of independent mechanisms, including other receptors involved in regulating oestrogen levels. Li et al. discovered a multi-step molecular pathway of cell death induced by high concentrations of oestrogens (5–10 μM) 17-β-oestradiol, progesterone, and oestrone, causing classical caspase-3-dependent apoptosis. They identified a new type of oestrogen receptor that triggers apoptosis, supporting an alternative mechanism of action for some clinically approved phosphodiesterase inhibitors. The authors also revealed a new mechanism of protein translation regulation, which may have potential implications for cancer control and inhibition of neurodegeneration [18,111].

A fundamental question arises regarding the newly discovered type of oestrogen receptor PDE3A, which may play a key role in regulating oestrogen homeostasis. It has been shown that oestrogens use a new oestrogen receptor to induce cell apoptosis. In addition, there are suspicions that oestrogen-like substances, which can interfere with endocrine homeostasis, might act through the newly discovered receptor. Could they induce apoptosis or cause oxidative stress in this mechanism?

This fascinating connection between oestrogen and cortisol raises many intriguing questions. Both the hormones oestrogen and cortisol play crucial roles in the human body. Oestrogen, primarily known as the female hormone, and cortisol, referred to as the stress hormone, seem to have similarities and differences, making understanding their interaction incredibly important.

Studies suggest a complex network of connections between these two hormones. For instance, oestrogen and cortisol are regulated by various feedback mechanisms that control their production and actions in the body. Additionally, both hormones influence many everyday processes, such as regulating metabolism, immune response, cognitive functions, and mood [246].

However, prolonged elevated levels of cortisol, caused by chronic stress, can lead to dysregulation of many physiological processes, including oestrogen metabolism. This is significant because oestrogen plays a crucial role in the development and function of many tissues and in regulating the immune system [251]. Advanced genetic–molecular studies can help us understand the precise mechanisms through which stress may affect oestrogen homeostasis and lead to destabilisation. Understanding these signalling pathways may be crucial for understanding the role of chronic stress in the development of oestrogen-related diseases, including oestrogen-dependent cancers.

This way, delving into this fascinating interaction between oestrogen and cortisol allows a better understanding of the body’s physiology. It may also lead to the discovery of new therapies and preventive strategies for diseases associated with hormonal dysregulation.

The discussion on the impact of exogenous sources of oestrogen on health and the need for further molecular research is extremely important and complex. The search for new signalling pathways and receptors plays a key role in assessing and treating hormonal disorders. Understanding these mechanisms is crucial for science and medical practice, as it enables the diagnosis and therapy of reproductive problems.

Recently discovered kisspeptins, which regulate gonadotropin-releasing hormones (GnRH) in the hypothalamus by activating the kisspeptin receptor Gpr54, are particularly interesting in the context of reproductive function. Disruption or loss of kisspeptin signalling leads to hypogonadotropic hypogonadism [252]. The impact of exogenous compounds disrupting hormonal homeostasis is a significant aspect of the analysis of sexual and hormonal disorders. The activity of kisspeptins in Leydig cells of the testes, controlling the progression of germ cells and sperm function, confirms the necessity of further research in this field [252].

One such exogenous compound is BPA, which has widely documented epigenetic effects on reproduction in both men and women. BPA disrupts spermatogenesis and sperm quality and induces transgenerational effects on the reproductive abilities of offspring. BPA affects ovarian function, embryo development, and gamete quality in women, which is crucial for successful in vivo and in vitro fertilisation and developing therapies [253].

Further molecular research is necessary to understand better how BPA affects health. Pregnancies, perinatal, and neonatal periods are critical because BPA can disrupt neuronal differentiation and synaptic plasticity. It induces neuroinflammation, neurodegeneration, and cognitive impairments [254]. Studies show that BPA affects the proliferation of peripheral blood mononuclear cells, causes chromosomal aberrations, and causes DNA damage. Moreover, it alters protein expression and immune surveillance functions, which can lead to the risk of cancer and neurodegenerative diseases [255].

Monitoring free and conjugated BPA levels in plasma and adipose tissue and conducting morph-functional in the liver is crucial for understanding the long-term effects of exposure to this compound. Higher levels of BPA were found in young rats exposed to BPA, suggesting that chronic exposure to low doses of BPA may increase the risk of diseases in adulthood [256].

Similar research must be applied to the endocannabinoid system (ECS), which integrates with many other cell signalling systems, including the oestrogen system (ES). Oestrogens, through the activation of oestrogen receptors, regulate growth, differentiation, and many other functions across a wide range of tissues in both men and women. Therefore, understanding the potential interactions between ECS and ES at the central and peripheral levels is crucial for further research in neuroendocrinology, reproduction, and oncology [257].

Monitoring exogenous sources of oestrogen and conducting molecular research is essential for a comprehensive understanding of their impact on health. A multifaceted approach, encompassing biochemical analyses and molecular studies, is key to effectively diagnosing and treating hormonal disorders and related diseases.

Based on the presented knowledge, the ambivalence of oestrogens as hormones essential for life but simultaneously potentially dangerous is highlighted. Further research into the mechanisms of oestrogen action and its impact on the body is necessary to understand better and control its effects. A balanced approach to hormone therapy and strict monitoring of oestrogen levels can help minimise the risks associated with their excess or deficiency while ensuring the health benefits resulting from their proper action.

## Figures and Tables

**Figure 1 ijms-25-08167-f001:**
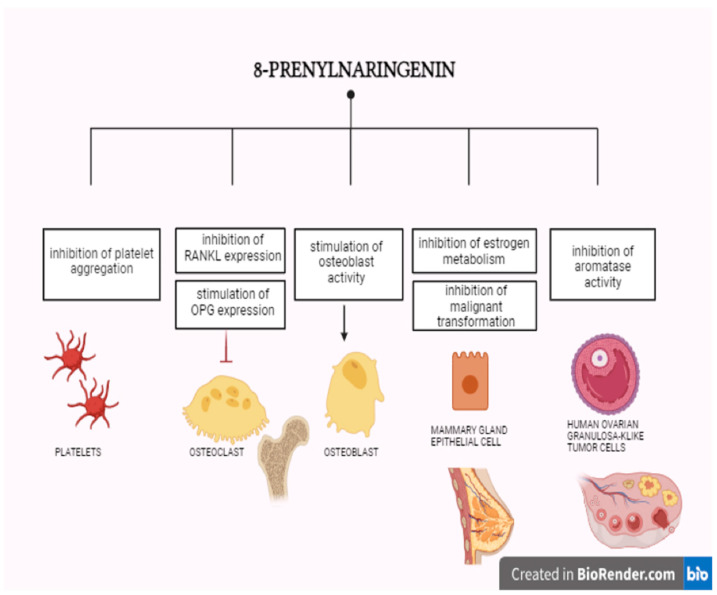
Effects of 8-prenylnaringenin based on in vitro studies [created in BioRender.com].

**Figure 2 ijms-25-08167-f002:**
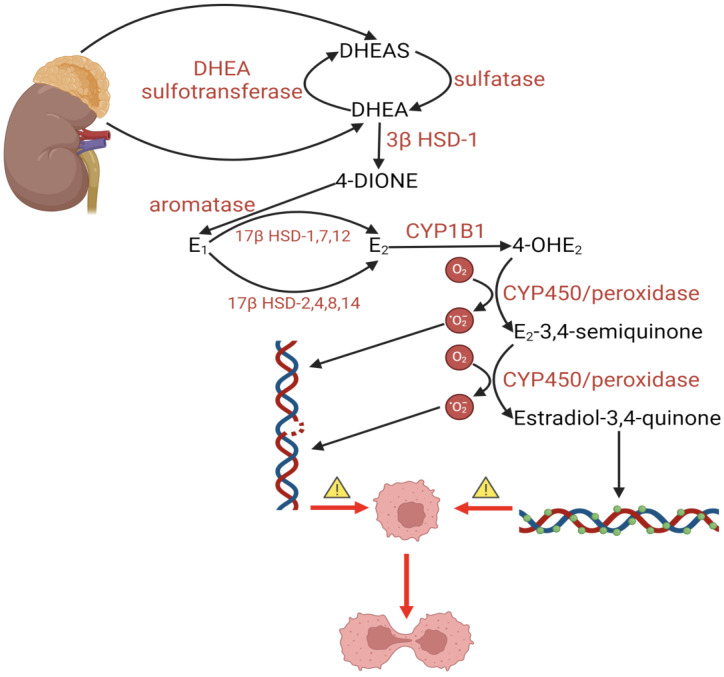
The transformations of DHEA from its synthesis in the adrenal glands to the final products promote the proliferation of cancer cells [created in BioRender.com].

**Figure 3 ijms-25-08167-f003:**
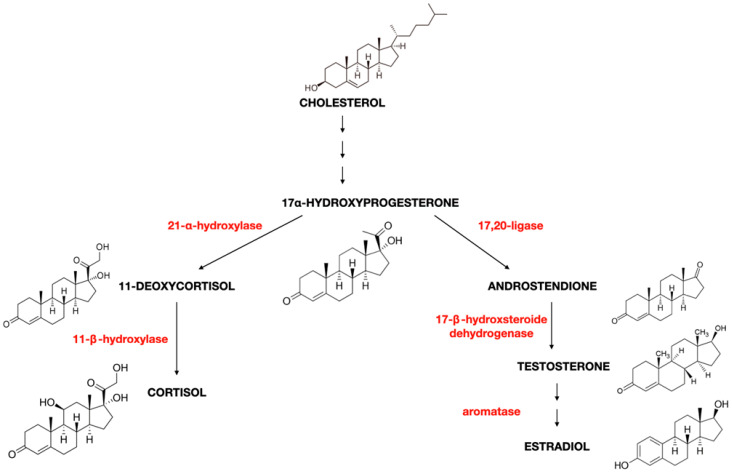
The metabolic pathway of oestrogen and cortisol biosynthesis from the precursor 17-α-Hydroxyprogesterone.

**Table 1 ijms-25-08167-t001:** Functions and regulation of aromatase.

Process	Description	Tissue	Function of Aromatase	Regulation of Expression
Conversion of androstenedione to oestrone	Aromatase converts androstenedione to oestrone through hydroxylation and dehydration.	Ovaries	Main site of oestrogen synthesis in premenopausal women. In ovarian follicle granulosa cells, aromatase converts androgens from theca cells to oestrogens.	Expression stimulated by LH ^1^
Conversion of testosterone to oestradiol	Testosterone is converted to oestradiol also through hydroxylation and dehydration.	Adipose tissue	Main site of oestrogen production in postmenopausal women. In adipose tissue, aromatase converts excess androgens to oestrogens, crucial for maintaining oestrogen levels post menopause.	Expression regulated by insulin and cytokines
Reduction of oestrone to oestradiol	Estrone can be reduced to oestradiol by the enzyme 17β-HSD. ^2^	Placenta	During pregnancy, placental aromatase synthesises oestrogens necessary for maintaining pregnancy and proper foetal development.	Regulated by various hormones and growth factors
	Aromatase is also present in other tissues where locally synthesised oestrogens may have paracrine and autocrine functions.	Brain, bones, muscles, skin		Regulated by local signals

^1^ luteinising hormone (LH). ^2^ 17β-HSD (17β-hydroxysteroid dehydrogenase).

**Table 2 ijms-25-08167-t002:** The comparison of oestrogen’s effects on the physiology of women and men [35,36,37,38,39,40].

Aspect	Women	Men
Bone health and growth	Oestrogen is crucial for bone maturation and density. Deficiency leads to delayed skeletal maturation and osteoporosis.	Similar effects as in women: tall stature without a pubertal growth spurt, delayed skeletal maturation, and severe osteopenia. Bone density improves with oestrogen replacement.
Reproductive system	Essential for reproductive function; influences ovarian and uterine health. Oestrogen receptor α (ERα) plays a significant role in fertility and uterine function.	Leads to hypergonadotropism, macro-orchidism, and increased serum testosterone when deficient. Oestrogen is crucial for proper testicular function and spermatogenesis.
Cardiovascular health	Protective against cardiovascular diseases; affects lipid and carbohydrate metabolism.	Deficiency can lead to premature coronary atherosclerosis due to its impact on lipid and carbohydrate metabolism.
Skeletal system	Important for growth plate chondrocytes; accrual and the maintenance of bone mass and density.	Oestrogen deficiency in men has a significant impact on skeletal growth and bone mass, similar to women.
Metabolism	Influences carbohydrate and lipid metabolism, contributing to the regulation of body weight and composition.	Oestrogen plays a role in metabolic processes, and its deficiency can negatively impact carbohydrate and lipid metabolism.
Immune system	Oestrogen impacts the immune system, contributing to immune response regulation.	Similar effects on the immune system as in women, though specific impacts might vary.
Endocrine function	Affects neuroendocrine system and hormonal balance. Plays a role in the timing of pubertal onset.	Oestrogen influences neuroendocrine function and hormonal regulation.
Developmental effects	Critical for normal development and function of the reproductive organs and secondary sexual characteristics.	Essential for normal sexual development and reproductive function; deficiency can lead to eunuchoid proportions and other developmental issues.

**Table 4 ijms-25-08167-t004:** Changes in the expression of selected gene variants during the induction and metastasis of carcinogenesis under the influence of phthalates: a group of endocrine-disrupting compounds capable of activating the ER.

Gene:	Regulation: Up/Down ↑/↓	Cancerogenesis Process
*CEACAM5*	↑	Protection from apoptosis, adhesion, and inhibition of differentiation [159]
*CYP1A1*	↑	AMPK inhibition: excessive proliferation and prolonged viability of cancer cells [160]
*DDIT4*	↑	mTOR activity suppression: excessive proliferation and apoptosis disorders [134,142]
*IER3*	↑	Apoptosis disorders [145]
*KLHL24*	↑	Overexpression of KLHL24 [136]
*SLC7A5*	↑	Provides the tumour with access to amino acid development and proliferation and activates the mTORC1 pathway [152]
*SLC7A11*	↑	Protection of cancer cells from oxygen radicals; Delivery of glucose and glutamine to cancer cells [15]
*STC2*	↑	Increased proliferation and viability; Regulation of the MAPK pathway [154]
*ADORA1*	↓	Proliferation inhibition (positive effect) and inhibition of ERalpha transcriptional activity [140]
*CCNA2*	↓	Cell cycle arrest and induction of apoptosis [161]
*CDK1*	↓	Cell cycle arrest [162]
*FKBP4*	↓	Proliferation and growth inhibition (positive effect) [163]
*PGR*	↓	Increased risk of recurrence [164,165]
*SFPQ*	↓	Excessive proliferation, tumour growth [166]
*TFAP2C*	↓	Excessive proliferation, tumour growth [167]

**Table 5 ijms-25-08167-t005:** Changes in the expression of selected gene variants during the induction and metastasis of carcinogenesis under the influence of endocrine-disrupting compounds with a similar mechanism of action to oestradiol-organic esters of phosphoric acid (OPEs).

Gene:	Regulation: Up/Down ↑/↓	Cancerogenesis Process
*CCNG2*	↑	Degradation of CDK2, inhibition of proliferation, and cell cycle arrest [136]
*CD55*	↑	More durable and regenerable cancer cells, proliferation, angiogenesis, and immune system evasion [168]
*CEACAM5*	↑	Avoidance of apoptosis, adhesion, and inhibition of differentiation [159]
*CYP1A1*	↑	AMPK inhibition: excessive proliferation and prolonged viability of cancer cells [160]
*DDIT4*	↑	mTOR activity suppression: excessive proliferation and apoptosis disorders [134,142]
*FOSL2*	↑	Participates in activating the metastasis cascade [143]
*HSPA13*	↑	TANK stabilisation, proliferation, and migration [144]
*IGF1R*	↑	Inhibition of apoptosis, proliferation, and enhanced ER activation [146,147]
*KLHL24*	↑	Cancer progression via influencing cellular proliferation and survival mechanisms [136]
*MAP1B*	↑	Inhibition of apoptosis (via p53) [169]
*RUNX2*	↑	Bone metastasis and impaired function and development of osteoblasts [149]
*SLC7A2*	↑	Inflammation and oxidative stress caused by increased NO synthesis, but inhibited invasion and migration [150,151]
*SLC7A5*	↑	Provides the tumour with access to amino acid development and proliferation and activates the mTORC1 pathway [152]
*SLC7A11*	↑	Protection of cancer cells from oxygen radicals and delivery of glucose and glutamine to cancer cells [153]
*STC2*	↑	Increased proliferation and viability, and regulation of the MAPK pathway [154]
*VEGFA*	↑	Angiogenesis and increased vascular permeability [170]
*CCNE1*	↓	Cell cycle and proliferation inhibition, and reduced cell viability [171]
*CEBPA*	↓	Suppressor gene silencing: accelerated proliferation [172]
*KRT19*	↓	Increased invasiveness, migration, and proliferation of cancer cells [173]
*PRKCD*	↓	Accelerated development of cancer cells [158]
*SFPQ*	↓	Excessive proliferation and tumour growth [166]
*TNFAIP2*	↓	Migration and invasion, proliferation, and angiogenesis [174]

**Table 6 ijms-25-08167-t006:** EC50 values, RBA for ER-α and ER-β, and occurrence of selected compounds from the phytoestrogen group and E2 [2,5,6,7,8].

Chemical Compound	EC_50_	RBA^β^ [%]	RBA^α^ [%]	Occurrence
8-Prenylnaringenin	4	6.5	19.5	Common hop
Genistein	200	79	6	Soybean
Coumestrol	30	35	22	Red clover
E2	0.8	100	100	Endogenous oestrogen

**Table 7 ijms-25-08167-t007:** Correlation between various types of oestrogen-using therapies and risk of breast cancer among women.

Type of Therapy	(n)	Breast Cancer Risk (Compared to Control Group)	Breast Cancer Risk (Related to Specific Conditions of the Therapy)	Breast Cancer Risk (Related to Substance Used during Treatment)
Hormonal contraception	1,797,932	1.2 (RR)	≥5 years of usage: 1.24 >10 years of usage: 1.38	Oestradiol valerate and dienogest: 1.62 Levonogestrel: 1.93 [228]
Hormonal replacement therapy	696,084	1.25 (HR)	<2 years of treatment: 1.08 2–5 years of treatment: 1.33 >5 years of treatment: 1.72	No data [226]
	98,611	1.21 (OR)	Oestrogen–progesterone: <1 year of treatment: 1.05 1–4 years of treatment: 1.34 ≥5 years of treatment: 1.57	Oestrogen– progesterone: 1.26 Oestrogen: 1.06 [227]
In vitro fertilisation (IVF)	25,108	1.01 (SIR)	≥7 IVF cycles: 0.55 (compared with 1–2 IVF cycle group) <4 oocytes collected: 0.77 (compared with group with ≥4 oocytes collected) [231]	-
	21,025	1.1 (HR)	Age at start of IVF: 24 years old: 1.56 32 years old: 1.16 36 years old: 1.00 [229]	
	7162	-	Age at start of IVF: >30 years old: 1.24 (RR) [230]	
Pharmalogical infertility treatment	1856	1.13 (OR)	<6 months of treatment: 0.4 ≥6 months of treatment: 1.7	Human menopausal gonadotropin: 2.25 Clomiphene citrate: 0.8 hCG and clomiphene citrate: 1.0 [233]
Gender-affirming hormone treatment	2260 trans women, 1229 trans men	Trans women: 0.3 (SIR, compared with cisgender women) 46.7 (SIR, compared with cisgender men) Trans men: 0.2 (SIR, compared with cisgender women) 58.9 (SIR, compared with cisgender men)	No data	No data [225]

**Table 8 ijms-25-08167-t008:** The differences between cortisol and oestrogen.

Characteristics/Properties	Cortisol	Oestrogen (Oestradiol)
Circadian rhythm	Exhibits a circadian rhythm with a peak within 45 min of waking, declining throughout the day and rising at night [236].	Exhibits a circadian rhythm with a morning peak and several ultradian harmonics throughout the 24 h period [241].
Differences between healthy and ill	The rhythm can differ between healthy and ill individuals; e.g., in SAD, the rhythm is delayed [237].	The rhythm remains relatively unchanged across menstrual cycle phases, except for the acrophase during the menstrual phase [241].
Phase studies	Numerous studies have examined the relationship of the cortisol circadian rhythm to other physiological processes [237,238,239].	Few studies on the circadian rhythm of oestradiol alone; some studies compare oestradiol and cortisol rhythms [241,242].
Phases in diseases	In diseases such as depression, the cortisol rhythm may be advanced or delayed [238].	The oestradiol rhythm does not phase-shift in preterm labour compared to term labour [242].
Relationship with other hormones	Exhibits various phase differences in relation to hormones such as TSH, prolactin, and growth hormones [237,238,239].	Studies suggest possible decoupling of oestradiol and cortisol rhythms in depression [241].
Influence of the menstrual cycle	May vary depending on the phase of the menstrual cycle [243,244].	The character of the oestradiol rhythm is relatively unaffected by the menstrual cycle, except for the acrophase [241].
Other	Phase changes in cortisol may indicate differences in health and disease states [237,238,239,240].	Few studies on optimal phase differences between oestradiol and other hormonal rhythms [245].

## Data Availability

Data sharing is not applicable.

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
