# Peer review of "The Dual Faces of Oestrogen: The Impact of Exogenous Oestrogen on the Physiological and Pathophysiological Functions of Tissues and Organs"

_ijms, 2024, doi:10.3390/ijms25158167_

Round 1

Reviewer 1 Report

Comments and Suggestions for Authors

Journal: International Journal of Moleular Sciences

Manuscript number#:

Article type: Review

Title: The Dual Faces of Estrogen: Impact of Exogenous Estrogen on Physiological and Pathophysiological Functions of Tissues and Organs

This paper explained the ambivalence of estrogen in detail with various examples and pathways. It has also pointed out the need for further research on estrogen. All the contents are meaningful and interesting, but I think some additional revisions are needed.

Major comments

1)      It would be good to have more information on estrogens other than the ones primarily mentioned in this article.

2)      I recommend that you write more about additional estrogen-related experiments.

Minor comments:

1)      Check and fix the spacing in all the content.

2)      Make sure the arrows have the same shape and size.

3)      It would be nice to make tables and figures more visible to make them easier to read.

Comments on the Quality of English Language

Journal: International Journal of Moleular Sciences

Manuscript number#:

Article type: Review

Title: The Dual Faces of Estrogen: Impact of Exogenous Estrogen on Physiological and Pathophysiological Functions of Tissues and Organs

This paper explained the ambivalence of estrogen in detail with various examples and pathways. It has also pointed out the need for further research on estrogen. All the contents are meaningful and interesting, but I think some additional revisions are needed.

Major comments

1)      It would be good to have more information on estrogens other than the ones primarily mentioned in this article.

2)      I recommend that you write more about additional estrogen-related experiments.

Minor comments:

1)      Check and fix the spacing in all the content.

2)      Make sure the arrows have the same shape and size.

3)      It would be nice to make tables and figures more visible to make them easier to read.

Author Response

Dear Reviewer,

We sincerely thank you for the valuable comments, the implementation of which will improve the substantive and scientific value of the work. We accept all the comments and have addressed them, making the appropriate revisions according to your suggestions, as follows:

Major Revisions:

  1. It would be good to have more information on estrogens other than the ones primarily mentioned in this article.

Thank you very much for this valuable comment. We have restructured the entire article, adding several subsections concerning the comprehensive role of estrogen in both physiology and pathophysiology, not only in women but also in men. All newly added sections are marked in green in the manuscript. The revised and restructured manuscript is attached to this message.

  1. I recommend that you write more about additional estrogen-related experiments.

Thank you for this comment. The experiments have been included in the newly added section, presenting additional functions of estrogens. These include in vitro and in vivo experiments, as well as clinical observations and studies.

Minor Revisions:

  1. Check and fix the spacing in all the content.

Thank you for the comment. We have corrected the spacing in the content. In cases of graphic editing and text arrangement issues, which we sometimes cannot control due to using different text editors, we rely on the assistance of the Editorial team.

  1. Make sure the arrows have the same shape and size.

Thank you for the comment. We have tried to ensure that the arrows have the same size and shape in our text editor.

  1. It would be nice to make tables and figures more visible to make them easier to read.

Thank you for the comment. We have highlighted the presence of Tables and Figures in bold, both in the text and in the captions above or below the Tables/Figures.

Kind regards,

Joanna Bartkowiak-Wieczorek

Reviewer 2 Report

Comments and Suggestions for Authors

The submitted review article aims at summarizing the “The Dual Faces of Estrogen: Impact of Exogenous Estrogen on Physiological and Pathophysiological Functions of Tissues and  Organs”. The selected topic is very large and the authoirs faild the goal to provide a comprehensive review. The manuscript is primarily focused on data in female reproduction and cancers, and mechanisms related to classical nucler receptors only are included . References in males, data in brain and other information deserving attenction are cited in discussion only. Hence, the title does not parallel the content of the manuscript at all. Furthermore, the organization of the manuscript is poor and the manuscript requires   rewriting,  inclusion of new paragrapohs, correction of wrong infirmation, and presentation of data with a clear rationale.

In particular:

·       The authors do not consider the membrane estrogen receptor and the estrogen related receptors: both classes are critical in estrogen signalling in health and disease and some of them specifically interact with estrogen-related ECDs.

·       Please, do not link estrogens to female (reproduction and cancer) only. Add information on the activity of estrogen in  male (reproduction and other), include data on brain  or modify the title accordingly (i.e., include the word  “female”)

·       Prior the description of ECDs, provide a description of the role of endogenous estrogens and related receptors

·       The organization of par 2 is questionable, due to the lack of a rationale. The paragraph is a mixture of information, not linked each other and sometime out of the context. Just to make a few examples: 90” The ligands of BPA”: BPA is not a receptor having ligands;

93-96 “Consequently, they affect the increased expression of 14 genes (CEL SR2, FOSL 2, JUN, HSPA 13, IER3, ADORA1, DDIT4, IGF1R, PGR, RUNX2, SLC7A11, SLC7A2, SLC7A5, STC2) and  decreased expression of 3 genes (BCAS3, PHF19, PRKCD) [22]. ]. The impact on these genes and their significance is summarized in [Table 1].” This sentence is completely out the context. The list of genes activated or repressed by ECDs exposure is really long (and also arrays are available). The provided list is referred to one study carried out in MCF7 cells only; thisd pòount has to be specifyed.

In my opinion this paragraph need rewriting and division in specific subparagraphs (e.g., bisphenols, types source, source and signaling; bisphenols in cancer; bisphenols in reproduction)

·       Use Italic cont for the name of genes and Mrna

·       Define abbreviations at the first appearencve within the main text and use consistently all over the text.

·       Reference list is not formatted accordingly the journal style

·       Please consider the following references: PMID: 27593959, PMID: 28990514, PMID: 31362658, PMID: 34575829, PMID: 33478092, PMID: 32334263

Comments on the Quality of English Language

Moderate editing of English language required

Author Response

Dear Reviewer,

Thank you for your detailed and constructive feedback on our manuscript. We have carefully addressed each of your comments and have made significant revisions to improve the quality and clarity of our work. Below are our responses to your specific concerns:

Reviewer Comment: "The submitted review article aims at summarizing the 'The Dual Faces of Estrogen: Impact of Exogenous Estrogen on Physiological and Pathophysiological Functions of Tissues and Organs'. The selected topic is very large and the authors failed the goal to provide a comprehensive review. The manuscript is primarily focused on data in female reproduction and cancers, and mechanisms related to classical nuclear receptors only are included. References in males, data in brain and other information deserving attention are cited in discussion only. Hence, the title does not parallel the content of the manuscript at all. Furthermore, the organization of the manuscript is poor and the manuscript requires rewriting, inclusion of new paragraphs, correction of wrong information, and presentation of data with a clear rationale."

Our Response: Thank you for this observation. We have extensively revised the manuscript to provide a more comprehensive review. We have added multiple new sections discussing the physiological role of estrogen in males, the impact of estrogen on the brain, and other relevant systems. The title has also been adjusted to reflect the expanded content of the manuscript better. Additionally, we have reorganised the manuscript for improved clarity and coherence.

Reviewer Comment: "The authors do not consider the membrane estrogen receptor and the estrogen related receptors: both classes are critical in estrogen signalling in health and disease and some of them specifically interact with estrogen-related ECDs."

Our Response: We have now included a detailed subsection on membrane and estrogen-related receptors, highlighting their critical roles in estrogen signalling in health and disease.

Reviewer Comment: "Please, do not link estrogens to female (reproduction and cancer) only. Add information on the activity of estrogen in male (reproduction and other), include data on brain or modify the title accordingly (i.e., include the word 'female')."

Our Response: We have incorporated additional sections that discuss the roles of estrogen in male reproduction and other physiological processes. Data on the impact of estrogen on the brain have also been included, and the title has been modified to represent the scope of the review more accurately.

Reviewer Comment: "Prior to the description of ECDs, provide a description of the role of endogenous estrogens and related receptors."

Our Response: We have added a separate subsection describing the role of endogenous estrogens and their associated receptors, providing a clearer context for discussing ECDs.

Reviewer Comment: "The organization of par 2 is questionable, due to the lack of a rationale. The paragraph is a mixture of information, not linked each other and sometime out of the context. Just to make a few examples: 90” The ligands of BPA”: BPA is not a receptor having ligands; 93-96 'Consequently, they affect the increased expression of 14 genes (CEL SR2, FOSL 2, JUN, HSPA 13, IER3, ADORA1, DDIT4, IGF1R, PGR, RUNX2, SLC7A11, SLC7A2, SLC7A5, STC2) and decreased expression of 3 genes (BCAS3, PHF19, PRKCD) [22]. ]. The impact on these genes and their significance is summarized in [Table 1].' This sentence is completely out the context. The list of genes activated or repressed by ECDs exposure is really long (and also arrays are available). The provided list is referred to one study carried out in MCF7 cells only; this point has to be specified."

Our Response: We have corrected the stylistic error regarding BPA ligands and removed the extraneous sentence. We have clarified that the list of genes pertains specifically to studies conducted on MCF7 cells. The paragraph has been reorganised and divided into specific subparagraphs for bisphenols, their sources, signalling, roles in cancer, and reproduction.

Reviewer Comment: "In my opinion this paragraph needs rewriting and division into specific subparagraphs (e.g., bisphenols, types source, source and signaling; bisphenols in cancer; bisphenols in reproduction)."

Our Response: We have rewritten and restructured the paragraph as suggested, dividing it into clear subparagraphs to enhance readability and logical flow.

Reviewer Comment: "Use Italic font for the name of genes and mRNA."

Our Response: We have formatted the names of genes and mRNA in italic font throughout the manuscript.

Reviewer Comment: "Define abbreviations at the first appearance within the main text and use consistently all over the text."

Our Response: We have defined all abbreviations at their first appearance and ensured consistent use throughout the manuscript.

Reviewer Comment: "Reference list is not formatted accordingly the journal style."

Our Response: We have reformatted the reference list to comply with the journal's style requirements.

Reviewer Comment: "Please consider the following references: PMID: 27593959, PMID: 28990514, PMID: 31362658, PMID: 34575829, PMID: 33478092, PMID: 32334263."

Our Response: We have included all the suggested references in the revised manuscript.

Comments on the Quality of English Language: "Moderate editing of English language required."

Our Response: We have undertaken a thorough revision to improve the quality of English throughout the manuscript.

Attached is the revised manuscript. The added sections and changes are highlighted in green to facilitate your review. We appreciate your thorough critique and believe that these revisions have significantly enhanced the manuscript.

Sincerely,

Joanna Bartkowiak-Wieczorek

Round 2

Reviewer 2 Report

Comments and Suggestions for Authors

The authors substantially revised their manuscript that in the present version is better organized and structured. They included new information on estrogens in not reproductive (female) tissues, corrected several mistakes, better structured their manuscript; they only considered the suggested references for discussion (most of them are master review articles in the field usefull in different section of the submitted manuscript). Taken together the manuscript has sufficient quality fior publication. 

I only have few minor quriess:

Table 2 requires references

345 [Mehra et al., 2005] insert the number of reference

417 [92 Khalaj et al., 2013]. Delete Khalaj et al., 2013

578 [121 Teng et al., 2013]. Delete Teng et al., 2013

730 [191 Veronese, 2016]. Delete Veronese, 2016

What is the meaning of asterisks in table 4 and 5?

Be consistent with the use of abbreviations (e.g., Erα or Er-α?)

Define abbreviations only at the first appearence within the main text and use consistently all over the main text; take care to use Italic font for the name of genes

Comments on the Quality of English Language

Minor editing of English language required

Author Response

Dear Reviewer,

Thank you for your detailed review and valuable comments. We have made the indicated corrections as described below:

  1. Table 2 requires references – We have added references to Table 2. Thank you for pointing this out.

  2. 345 [Mehra et al., 2005] insert the number of reference – We have added the reference number. Thank you for this suggestion.

  3. 417 [92 Khalaj et al., 2013]. Delete Khalaj et al., 2013 – We have deleted the mention of Khalaj et al., 2013. Thank you for this comment.

  4. 578 [121 Teng et al., 2013]. Delete Teng et al., 2013 – We have deleted the mention of Teng et al., 2013. Thank you for this comment.

  5. 730 [191 Veronese, 2016]. Delete Veronese, 2016 – We have deleted the mention of Veronese, 2016. Thank you for this comment.

  6. What is the meaning of asterisks in table 4 and 5? – We have removed the asterisks as they were an editorial error. Thank you for pointing this out.

  7. Be consistent with the use of abbreviations (e.g., Erα or Er-α?) – We have ensured consistency in the use of abbreviations throughout the text. Thank you for this suggestion.

  8. Define abbreviations only at the first appearance within the main text and use consistently all over the main text; take care to use Italic font for the name of genes – We have revised the text to define abbreviations only at their first appearance and have used italic font for gene names throughout the manuscript. Thank you for this valuable suggestion.

  9. Minor editing of English language required – We have made the necessary grammatical and stylistic corrections. All changes are highlighted in blue in the attached draft. Thank you for this comment.

Once again, we appreciate all your comments. Your thorough review has significantly improved the quality of our manuscript.

Sincerely,

Joanna Bartkowiak-Wieczorek
